



# The Green Ocean: Precipitation Insights from the GoAmazon2014/5 Experiment

Die Wang[1], Scott E. Giangrande[1], Mary Jane Bartholomew[1], Joseph Hardin[2], Zhe Feng[2], Ryan Thalman[3], and Luiz A. T. Machado[4]

[1]Environmental and Climate Sciences Department, Brookhaven National Laboratory, Upton, NY, USA
[2]Pacific Northwest National Laboratory, Richland, WA
[3]Department of Chemistry, Snow College, Richfield, UT, USA
[4]National Institute for Space Research, Sao Jose dos Campos, Brazil

*Correspondence to:* D. Wang (diewang@bnl.gov)

**Abstract.** This study summarizes the precipitation properties collected during the GoAmazon2014/5 campaign near Manaus in central Amazonia, Brazil. Precipitation breakdowns, summary radar rainfall relationships and self-consistency concepts from a coupled disdrometer and radar wind profiler measurements are presented. The properties of Amazon cumulus and associated stratiform precipitation are discussed, including segregations according to seasonal (Wet/Dry regime) variability, cloud echo-top height and possible aerosol influences on the apparent oceanic characteristics of the precipitation drop size distributions. Overall, we observe that the Amazon precipitation straddles behaviors found during previous U.S. Department of Energy Atmospheric Radiation Measurements program (ARM) tropical deployments, with distributions favoring higher concentrations of smaller drops than ARM continental examples. Oceanic type precipitation characteristics are predominantly observed during the Amazon Wet seasons. An exploration of the controls on Wet season precipitation properties reveals that wind direction, as compared with other standard radiosonde thermodynamic parameters or aerosol count/regime classifications performed at the ARM site, provides a good indicator for those Wet season Amazon events having an oceanic character for their precipitation drop size distributions.

## 1 Introduction

Global climate models (GCMs) continuously improve to overcome deficiencies in climate predictions associated with cloud and precipitation processes (e.g., Klein and Genio, 2006; Del Genio, 2012). Following suit, observational studies serve to better inform GCM and cloud resolving model (CRM) activities by providing the physical understanding for more diverse, climatically significant global cloud conditions and their associated feedbacks. Cumulus to deeper convective clouds are associated with high impact weather events and act as the engine of the global circulation. Subsequently, model treatments of convection have profound impact on weather and climate simulations. For practical reasons, the evaluation of cumulus treatments has of-





ten stressed comparisons against larger-scale, longer-term precipitation properties; for example, accumulated rainfall products from ground or space-borne instruments (e.g., Hou et al., 2014). Thus, improving precipitation measurements (i.e., those that better reflect the natural variability of clouds under diverse forcing conditions) has traditionally supported improved convective treatments. Nevertheless, improving model capabilities are introducing new challenges that motivate multi-scale, multi-sensor

observations to better constrain cloud microphysics and dynamics closer to the process levels future GCMs attempt to represent (e.g., Mather and Voyles, 2013; Donner et al., 2016).

Recently, the Observations and Modeling of the Green Ocean Amazon (GoAmazon2014/5) Experiment was motivated by demands to gain a better understanding of aerosol, cloud and precipitation interactions on climate and the global circulation (Martin et al., 2016, 2017). The inability of GCMs to adequately represent cumulus clouds and precipitation over the Amazon

highlights one example of the observational needs for future improvement to GCM cloud parameterizations and the larger-scale circulation connections therein (e.g., Richter and Xie, 2008; Nobre et al., 2009; Yin et al., 2013). A low-level barrier when developing useful observational constraints is the shortage of long-term datasets for reference within tropical regions such as the Amazon basin. Geophysical retrievals of interest for precipitation studies include radar-based rainfall estimation, but even basic radar preprocessing improvements for dual-polarization quantities can be critical for future studies utilizing forward

radar model comparisons. Establishing boundaries for tropical measurement expectations and radar data quality concepts (e.g., Scarchilli et al., 1996, self-consistency methods) provides an immediate benefit when interpreting remote radar deployments including those from the Atmospheric Radiation Measurement (Ackerman and Stokes, 2003, ARM) Mobile Facility (Miller et al., 2016, AMF). Specifically, the Amazon basin offers a unique tropical perspective on the variability of precipitation, as it receives copious precipitation across diverse cloud conditions, including 'Wet' and 'Dry' seasonal variability interconnected

to large-scale shifts in the thermodynamic forcing and coupled local cloud-scale feedbacks (e.g., Machado et al., 2004; Cifelli et al., 2004; Li and Fu, 2004; Misra, 2008).

Although anchoring hydrological retrievals is of a practical significance, an interesting scientific perspective from previous Amazon precipitation studies are those promoting the Amazon as a 'Green Ocean'. This terminology references the unique regional characteristics observed from the convection that spans oceanic to continental cloud extremes in key attributes such

as updraft intensities and propensity for electrification. Specifically, Amazon clouds may initiate under clean (or lower) cloud condensation nuclei (CCN) conditions, over a pristine forest, but also experience a range of thermodynamical and aerosol forcing influences that promote changes in cloud properties including electrification, cloud droplet size distribution and precipitation changes, or enhanced updraft intensity (e.g., Williams et al., 2002; Cecchini et al., 2016; Giangrande et al., 2016b, 2017). As described by Williams et al. (2002), the prevalence of maritime convective cloud regimes over a large continent are

possibly still underappreciated in the convective spectrum and its intensity, especially given the propensity to identify deeper convection over the Amazon having electrification arguing continental convective characteristics.

To better understand the diversity of convective clouds as well as to constrain upcoming convective modeling activities from GoAmazon2014/5, it is informative to explore Amazon cumulus characteristics over this extended dataset. One motivation is to question, "When is the Green Ocean 'blue'?", towards identifying conditions under which precipitation sampled in the

Amazon basin adheres more towards oceanic, maritime to possible continental characteristics (e.g., Tokay and Short, 1996).





Previous ARM Tropical Western Pacific (e.g., Long et al., 2016, TWP) precipitation studies have helped identify practical thresholds and composite behaviors for convection, as well as the strengths/deficiencies for more practical convective cloud regime segregations under various larger-scale monsoonal and more oceanic cloud environments (e.g., Bringi et al., 2003; Giangrande et al., 2014a; Thompson et al., 2015). In addition to better addressing Amazon precipitation in the context of global variability, it is useful to assess whether traditional (or, practical) radar remote-sensing (including dual-polarization) quantities are sensitive to these shifts.

This study summarizes the precipitation properties collected by the ARM AMF during GoAmazon2014/5 at the 'T3' site located approximately 70 km to the west of Manaus in central Amazonia, Brazil ($3°12'46.70''$S, $60°35'53.0''$ W). The location samples both the local pristine atmosphere and possible effects of the Manaus, Brazil pollution plume. The T3 site was equipped to capture a continuous convective cloud and precipitation column characteristics from a reconfigured radar wind profiler coupled with a ground disdrometer. Section 2 describes the instrumentation, methods and sources for uncertainty in results presented by this study. Precipitation comparisons from the disdrometer using traditional drop size distribution (DSD) parameters and dual-polarization quantities useful for future hydrological applications are located in Section 3. Sections 4 and 5 discuss the properties of the Amazon cumulus convective and associated stratiform precipitation, including segregations according to seasonal (Wet/Dry regime) variability, cloud height and possible aerosol influences. A summary of the key findings and discussions about Amazon as a 'Green Ocean' are provided in the final section.

## 2   Dataset and Methodology

### 2.1   The ARM T3 Precipitation Dataset and Processing

Precipitation observations are obtained from two primary instruments, a second-generation PARSIVEL disdrometer (e.g., Löffler-Mang and Joss, 2000; Tokay et al., 2014; Bartholomew, 2014) and a reconfigured 1290 MHz radar wind profiler (Giangrande et al., 2013, 2016b; Coulter et al., 2009, RWP). The collocated sensors capture surface DSDs, as well as simultaneous profiles for the vertical velocity and reflectivity factor estimates through the depth of Amazon clouds. These instruments operated concurrently from September 2014 through December 2015, periods that captured one 'Wet' season (herein, the five months defined as December through April) and one 'Dry' season (herein, periods from June through September). Additional information on the AMF deployment, including details on the larger-scale thermodynamic sampling throughout the campaign and appropriateness for Wet/Dry regime separations, is found in the GoAmazon2014/5 overview by Giangrande et al. (2017).

PARSIVEL measurements such as estimated DSD parameters and additional derived radar quantities are determined using 5-min aggregation windows. This sampling reduces noisiness found in 1-minute DSD quantity retrievals, which is reduced further by selecting 5-minute DSDs having $R > 0.5$ mmhr$^{-1}$ and total drops $> 100$. In total, 3852 5-minute DSDs were collected during the GoAmazon2014/5 campaign, with 3087 associated with the collocated RWP observations. The total precipitation associated with the full set of DSD observations is 2597 mm, with 2511 mm associated with collocated RWP observations. Approximately 1500 mm were associated with convective precipitation (RWP classifications to be discussed in later sections). Processing for the disdrometer was performed using the open-source PyDSD code (Hardin, 2014), with standard corrections





(e.g., Tokay et al., 2013, 2014). Estimated quantities include the rainfall rate $R$ (mmhr$^{-1}$), rain water content LWC (gm$^{-3}$), measured median volume drop size $D_0$ (mm), the mass-weighted mean diameter $D_m$ (mm). Processing also includes additional parameters of a gamma-fit DSD assumed of the form $N(D) = N_0 D^\mu \exp(-\Lambda D)$, having equivalent volume diameter $D$ (mm) with number concentration $N_0$ (mm$^{-1}$m$^{-3}$), shape parameter $\mu$, and slope parameter $\Lambda$ (mm$^{-1}$) calculated following a method of

higher moments (second, fourth, and sixth moment, e.g., Cao and Zhang (2009)).

    Additional calculations for a normalized DSD intercept parameter $N_w$ have been adapted following Testud et al. (2001). These are required to investigate a DSD-based convective-stratiform partitioning scheme based on disdrometer observations (e.g., Bringi et al., 2002, 2003, 2009). Dual-polarization radar quantities including the radar reflectivity factor $Z$ (dBZ), differential reflectivity factor $Z_{DR}$ (dB), specific differential phase $K_{DP}$ (degkm$^{-1}$) and specific attenuation $A$ (dBkm$^{-1}$, horizontal

polarization) are estimated for liquid media at 20°C using a T-Matrix approach (e.g., Mishchenko et al., 1996). These estimates assume nonspherical drop shapes according to the relationship in Thurai et al. (2007) and standard drop canting assumptions for S-band (10-cm), C-band (5.45-cm) and X-band (3.16-cm) wavelengths.

    RWP measurement details have been summarized by several recent studies, with precipitation datasets available at high spatiotemporal resolution of approximately 15 seconds and 200 meters (e.g., Tridon et al., 2013; Giangrande et al., 2013). To

align with the 5-minute disdrometer measurements, an RWP profile from the mid-point of the 5-minute window are selected. RWP measurements are typically stable with respect to $Z$ calibration offsets, with measurements aligned to those from the surface disdrometer (typically viable to within 2 dB). Echo-top height (ETH) from the RWP is defined as the altitude where column $Z$ drops below 10 dBZ. Amazon RWP observations indicate this relative $Z$ altitude as the approximate height that mean convective cloud vertical velocity approaches 0 ms$^{-1}$. Vertical air velocity retrievals and echo classifications follow

techniques outlined by Giangrande et al. (2016b). For echo classifications, we first identify higher confidence convective and stratiform regions on the basis of column $Z$ signatures and/or so-called radar 'bright band' (melting-level) designations for longer wavelengths (e.g., Fabry and Zawadzki, 1995; Geerts and Dawei, 2004). In contrast to scanning radar-based echo designations (e.g., those typically use near surface $Z \simeq$ 40-45 dBZ thresholds, e.g., Steiner et al. (1995)), one RWP advantage is that columns exhibiting stronger vertical air velocity signatures help to further differentiate transitional or elevated convective

cells (e.g., instances of |VV| > 2 ms$^{-1}$).

### 2.2   The ARM T3 Aerosol Observations and Aerosol Regime Classification

Aerosol regime classifications are based on the combination of number concentration of particles condensation nuclei (CN), fraction of particles with diameters less than 70 nm, carbon monoxide CO and oxides of Nitrogen (NO$_y$) measurements at the T3 location using instrumentation as described in Thalman et al. (2017) and supplemental materials. The philosophy for

this aerosol classification that each aerosol measurement builds on the previous when establishing a background condition ('clean'), with additional support for 'polluted' conditions (e.g., urban, above this background condition) as well as 'biomass' burning conditions attributed on top of 'polluted' criteria. One advantage for using this classification is that this combination of measurements helps mitigate concerns that precipitation onset will mask ambient aerosol conditions (e.g., as in including an insoluble CO measurement). Because of the pronounced shift in aerosol properties seasonally, the methods sub-classify




background and polluted air mass types according to seasonal-specific windows. Classifications are available at 5-minute intervals that align with the available precipitation observations.

As summarized by Thalman et al. (2017), 'clean' conditions (typical background levels) exhibit values CN < 500 $\mathrm{cm}^{-3}$, CO < 0.14 ppm and $\mathrm{NO}_y$ < 1.5 ppbv during the Wet season. In contrast, background levels and associated classifications shift

towards values of CN $\simeq$ 1500 $\mathrm{cm}^{-3}$ (e.g., potentially factor of 3 or more difference in CN) for similar CO and $\mathrm{NO}_y$ thresholds during the Dry season (Transitional months fall between Wet/Dry characteristics). In this regard, the Dry season background conditions reflect regional biomass burning background levels that might otherwise be considered polluted conditions during typical Wet season months. For this study, sampling limitations during the GoAmazon2014/5 Dry season (lack of available collocated precipitation and aerosol measurements) requires our use of only Wet season observations to provide a more detailed

aerosol-cloud-precipitation investigation. Under Wet season conditions, 'polluted' regimes are those having CN > 500 $\mathrm{cm}^{-3}$. 'Biomass' regimes are considered a more stringent polluted classification, classified as those 'polluted' regimes also having CO > 0.14 ppm.

### 2.3   PARSIVEL Sampling and Rainfall Relationship Interpretation

Later sections document relationships between estimated radar quantities and the measured rainfall rate *R*. These quantities

carry instrument sampling considerations that include catchment uncertainties under convective conditions (e.g., Duchon and Essenberg, 2001), additional limitations when compared to collocated devices (e.g., Tokay et al., 2013; Park et al., 2017; Thurai et al., 2017), and/or processing assumptions when applying functional-form DSD parameter fits (e.g., Smith et al., 2009; Cao and Zhang, 2009). Radar quantities as estimated by disdrometer are also influenced by raindrop shape and additional assumptions introducing variability established in previous studies (e.g., Thurai et al., 2007). Given our comparisons between rainfall

accumulations with surface gauge measurements under typical storm intensities, as well as previous side-by-side performance testing of other PARSIVEL units, we do not anticipate estimated uncertainty falling outside the variability established by previous studies. However, reasonable instrument offsets for radar quantities such as *Z* may be on the order of 10-20 % or 1-2 dBZ.

An additional consideration when fitting rainfall relationships is the representativeness of this dataset, including challenges

when attempting to establish the significance of functional fits. We establish coefficients for conventional *R(Z)* relationships of the form $Z = aR^b$ using nonlinear least squares methods matched over the entire dataset (or subsets) of *Z-R* pairs. For lengthier datasets, it is informative to test variability in coefficients as related to modest samples drawn from the total population. Since consecutive DSD observations within precipitation events are non-independent and correlated in processes (e.g., Lee et al., 2009), it may be important to consider this impact of proper spacing between samples when ensuring a reliable relationship.

Figure 1 plots histograms for 'a'-coefficient values from various single parameter rainfall relationships (radar quantities estimated as in previous sections), assuming a fixed 'b'-coefficient as determined from our complete Amazon dataset for the S-band wavelength. This example highlights the sensitivity in the 'a'-coefficients as estimated from random half-dataset subsets to the complete dataset (vertical black line). As the radar wavelength decreases, the sensitivity depends on the radar quantity of interest. For example, the differences we observe for Amazon subsamples are typically to within 5 % of the mean dataset





'a'-coefficient value with respect to $R(Z)$. A deterioration to shorter wavelengths is found for $R(Z)$ relationships owing to the importance of larger diameters to $Z$ estimates and increased non-Rayleigh scattering. In contrast, 'a'-coefficients are found typically to within 2 % for $R(K_{DP})$ and $R(A)$ relations, with improved performance to shorter wavelengths (more immediate relationship between these quantities and rainfall rate). Such results help inform basic interpretations for the significant changes

in these radar relationship coefficients.

## 3   Summary Precipitation Results and Interpretation for Retrieval Methods

This section summarizes bulk precipitation properties, rainfall relationships, and basic dual-polarization radar connections for the GoAmazon2014/5 dataset. A summary of DSD parameter breakdowns for select quantities, filtered according to rainfall rate intervals, is located in Table 1. As one point of comparison to continental expectations, we include values obtained from

a year-long ARM Southern Great Plains (SGP) PARSIVEL2 deployment (November 2016 through October 2017), processed similar to the Amazon datasets. Within these narrowed rainfall rate intervals, the Amazon precipitation exhibits reduced median drop sizes and higher drop concentrations. This change is also reflected in lower $Z$ values and higher LWC for a similar $R$ as compared to SGP observations. Although the 5-minute Dry season samples are limited, rainfall rate breakdowns demonstrate the Dry season exhibits higher-relative $Z$ and median drop sizes (lower $N_w$ and LWC) as compared to Wet season observations.

Discrepancies between SGP and Amazon, as well as Wet/Dry separations, are most pronounced at the higher $R$ consistent with convective cores.

### 3.1   Single-parameter Dual-polarization Rainfall Relationships at S- C- and X-band

Figure 2 plots summary dataset scatterplots and overlaid dual-polarization relationship fits. A summary of matched rainfall coefficients is provided in Table 2. For these tables, 'b'-coefficients were fixed at a characteristic dataset value to facilitate

comparison across regime breakdowns. Figure 2 overlays the associated fitting (dashed lines) from SGP-Oklahoma to provide a continental reference. As a function of radar wavelength, the 'a'-coefficient values decrease as wavelength decreases (e.g., quantities more closely related to the rainfall rate). SGP dual-polarization relationships are consistent with previous studies (e.g., Giangrande et al., 2014b), providing confidence in the appropriateness of disdrometer processing. Note, minor discrepancies in SGP $R(Z)$ relations may be related to our filtering of large drops > 5 mm that disproportionately influence $Z$

measurements at sites including SGP wherein melting hail is more regularly observable. Moreover, SGP relations may carry higher 'a'-coefficients (thus, larger discrepancies with the Amazon relationships) than reported in our Table 2.

     Summary Amazon relationships follow a tropical expectation (more significant role for warm-rain processes, e.g., droplet growth via collision-coalescence), indicating higher concentrations of smaller drops. This is observed when having a smaller 'a'-coefficient than found for SGP $R(Z)$ relations, and larger 'a'-coefficients than found for SGP $R(K_{DP})$ and $R(A)$ relations (as

in Figure 2). These changes reflect a significant change when viewed compared to Amazon dataset sampling arguments found in Section 2. For Table 2, $R(A)$ relationships are also listed for multiple temperature assumptions, highlighting one explanation for modest variability when attempting to promote these relations for practical rainfall retrievals (e.g., Ryzhkov et al., 2014;



Giangrande et al., 2014b). As additional reference for dual-polarization radar processing and natural calibration concepts, self-consistency relationships between radar quantities $K_{DP}$, $Z$, and $Z_{DR}$ for the various wavelengths have been provided in Table 2. In comparison to continental SGP references for statistical DSD relationships (e.g., Ryzhkov et al., 2005), self-consistency coefficients in Table 2 reinforce the tropical character of the Amazon precipitation, again consistent with smaller median drop

sizing (e.g., reductions in $Z_{DR}$ or $K_{DP}$) to achieve similar estimates of $Z$.

     Rainfall relationships stratified according to Wet and Dry season conditions are also found in Table 2. The Wet season indicates lower 'a'-coefficients for $R(Z)$ and higher relative coefficients for $R(K_{DP})$ and $R(A)$ relations. One interpretation is that for similar $R$, the Wet season DSDs carry the more pronounced tropical precipitation characteristics. A similar trend is found with seasonal self-consistency relationship breakdowns. As before, most seasonal breakdowns are reflected as significant

changes in relationship coefficients when compared the sampling arguments in Section 2. An exception to this are $K_{DP}$-based rainfall relationships that appear least sensitive to this seasonal DSD variability at X-band (the shortest wavelength tested), possibly a reflection of non-Rayleigh influences on $K_{DP}$ (i.e., the presence/absence of larger drops) is less important.

     Finally, interpreting seasonal differences can be challenging without mentioning factors including storm intensity changes related to the larger-scale thermodynamic shifts that alter convective and congestus frequency, or mid-level moisture (e.g.,

during GoAmazon2014/5 as in Giangrande et al. (2017), cf. 2). The Dry season promotes storms that achieve higher rainfall rate $R$, but under convective environments favoring enhanced evaporation, cooling and subsidence less capable to sustain expansive stratiform processes. Wet and adjacent transitional month stratiform precipitation linked to aggregation and associated DSD evolution processes beneath the melting layer favors lower $N_w$, higher $D_0$ values for similar $Z$ (e.g., Giangrande et al., 2016a). The fraction of stratiform DSDs (count) to the total DSD observations in this dataset for the Wet season is 50.5 %. This fraction

decreases to approximately 30.1 % for the Dry season. Nevertheless, summary rainfall accumulations/properties skew heavily towards convective designations for all seasons, as reported in Table 3 and discussed further in Section 4.

### 3.2   Convective/Stratiform Regimes for Rainfall Relationships and DSD Properties

Isolating contributions from convective and stratiform DSDs is an initial step for improved rainfall estimates or possible model evaluation (e.g., Tokay and Short, 1996). Table 2 is segregated according to RWP-based convective/stratiform echo classifica-

tions. These RWP-based segregations will be further decomposed in Section 4, but for demonstration purposes are considered a reasonable benchmark when isolating bulk regime contributions. Cumulative contoured frequency altitude display histograms (e.g., Yuter and Houze, 1995, CFADs) with quantile values (median, 90th and 95th percentile) for RWP $Z$ and vertical velocity retrieval profiles are plotted in Figure 3. These histograms help establish these RWP classifications as reasonable; For example, convective columns (Figures 3a and 3c) have monotonically decreasing profiles and stronger vertical motions, whereas

stratiform (Figure 3b, 3d) columns emphasize pronounced radar 'bright band' - or, aggregation process signatures and weaker composite upwards vertical air velocity signatures (e.g., Fabry and Zawadzki, 1995). Given that checks for pronounced 'bright band' signatures are part of the echo classification, that these signatures are observed is not surprising. Inflation of mid-level downwards motions in stratiform regions is observed near the freezing level, originating from contamination within the melting layer on fall speed corrections (e.g., enhanced $Z$ from aggregation resulting in overestimates for ice fall speeds).





In terms of rainfall relationships in Table 2, convective relationships demonstrate higher coefficient values for $R(K_{DP})$ relations and smaller coefficients for $R(Z)$ relations. This shift is consistent with convection favoring high $N_w$, low $D_0$ for a similar $Z$ or $K_{DP}$. $R(A)$ relationships register as those least influenced by these separations (smallest coefficient shifts), followed by $K_{DP}$ relationships at the shorter wavelengths. This performance reflects on the closer relationship between $A$ and

$K_{DP}$ with rainfall rate, less influenced by the presence/absence of select larger drop sizes. As complementary examples for the Amazon datasets, Figure 4 plots the corresponding histograms for Amazon convective and stratiform DSDs in terms of $N_w$ (Figure 4a), LWC versus $R$ relationships (Figure 4b), and $D_0$ variations with $Z$ (Figure 4c). For these plots, convection is noted by red shadings and stratiform is plotted in blue contours. Convection demonstrates a broader distribution of $N_w$, LWC and other quantities of interest. Although there is substantial overlap with stratiform DSDs, convective DSDs exclusively cover

select (higher extreme) parameter spaces.

## 4    Amazon Precipitation Properties: Cumulative Dataset Characteristics

Convection-permitting models struggle to simultaneously capture convective and stratiform cloud processes, therefore model-observational comparisons often emphasize bulk cloud regime segregations and contingent performances to diagnose issues with cloud model treatments (e.g., Lang et al., 2003). Although there is no clear line separating convective and stratiform

processes (e.g., for identifying deficiencies in modeled precipitation, vertical air motions or heating profiles), bulk regime separations introduced for Section 3 are of practical use. Here, we assess how precipitation depictions from previous campaigns might be useful to constrain Amazon observations and the sensitivity of radar quantities to those changes.

Precipitating clouds identified by the RWP demonstrate a clear bimodal ETH distribution (Figure 5), and one that varies according to Amazon seasons (Figure 5a). The behaviors are consistent with freezing level (typically, around 5 km above

surface) and tropopause-level cloud-top expectations for tropical convection (Figure 5b, e.g., Johnson et al. (1999); Jensen and Del Genio (2006)). Note also that the RWP is not sensitive to cloud-sized particles, thus actual cloud top heights (as from collocated cloud radar referenecs) may extend 2 km or more above these heights. Sounding-based winds over the T3 site are predominantly easterly (mostly observed during the Dry season) to northeasterly (mostly, Wet season) (Figure 5c), with low-level $Z$ observations (Figure 5d) illustrating that Amazon cumulus are often linked to relatively modest values of $Z \simeq 35$ dBZ.

From a practical radar-based classification perspective that typically utilizes higher $Z \simeq 40\text{-}45$ dBZ thresholds, it follows that standard methods may necessitate additional texture, peakedness or similar ideas to properly identify Amazon convection (e.g., Steiner et al., 1995).

As documented by Giangrande et al. (2017, cf. 6 and 8), convection passing over T3 follows a diurnal cycle with peak cloud frequency around local 13-14 h. A shift in peak frequency to the later afternoon is found within the Dry season, whereas

Wet season deeper convection exhibits a secondary peak in cloud frequency (related to mesoscale convective systems) during the overnight hours. Congestus clouds (loosely, precipitating clouds having ETHs between 4.5 km and 9 km) demonstrate a similar diurnal pattern across all Amazon seasons. The frequency of all precipitating clouds (congestus and deeper) increases substantially for the Amazon Wet season. Of additional note, the precipitation originating from congestus or possible shallower





forms of tropical organized cloud systems (as defined solely on RWP-based ETH < 9 km in Chen and Liu (2015)) is nontrivial for this Amazon dataset (accumulations as reported in Table 3).

Figure 6 plots the frequency for observing various levels of vertical air motions (magnitude exceeding threshold as on Figure 6) within an RWP column as additional reference to the convective character of these clouds. Displays present these frequencies as a function of a lower-level RWP $Z$ ($\simeq 2$ km). To lower ranges of $Z$ ($< 35$ dBZ), we observe a stable percentage of columns having vertical air motions around $1$ ms$^{-1}$. This may be viewable as also the baseline uncertainty regarding RWP-based vertical velocity retrievals. As $Z$ increases above 35 dBZ, we observe a rapid increase in the frequency of stronger updrafts/downdrafts, indicative of the increasing contributions from convective clouds sharing these relative $Z$ levels. As $Z$ is stronger, the likelihood of sampling deeper clouds (and therefore additional chance to observe a stronger velocity in those column) also will increase as a function of $Z$. Results in Figure 6 also provide some guidance in convective/stratiform classification methods for scanning radars that use low-level Z thresholds (e.g., Steiner et al., 1995). Specifically, low-level $Z$ exceeding a 40 dBZ value (or higher) is a reasonable designation of convection in absence of vertical velocity measurements.

## 4.1 Disdrometer Convective-Stratiform Segregation: Alignment with RWP Signatures

Figure 7 plots a convective-stratiform regime segregation concept, with the solid line as reference to a DSD-based classification in following Bringi et al. (2003), herein BR. In this $N_w$ versus $D_0$ space, BR proposed that tropical maritime convective precipitation observed at Darwin, Australia falls to the right of the solid black line in Figure 7. In terms of thresholds, for this dataset the DSDs best aligned with falling on either side of the BR line correspond to those having a rainfall rate threshold of 13 mmhr$^{-1}$, or a $Z$ value of 40 dBZ. Figure 7 also overlays the contours the RWP-based classifications for convective (red colors) and stratiform (blue lines) precipitating columns. The ellipse on Figure 7 indicates the two-sigma confidence interval for those regions containing stratiform DSDs as based on the RWP classification.

RWP-based classifications indicate that substantial DSDs may be attributed to convective classifications left of this BR line. These are associated with the RWP identifying congestus or shallower convective cloud columns, as based on velocity signatures. However, the Amazon dataset supports bulk BR findings for deeper tropical convection in that precipitation to the right of the BR line is exclusive to convective designations. Since BR was developed using a Darwin monsoonal dataset, we anticipate that study included modest convective diversity into congestus clouds, maritime continental and deeper convective properties (those supporting additional graupel growth). Darwin may exhibit even more intense 'Break' (e.g., more continental characteristics) convective cell periods and associated DSD changes interspersed with maritime tropical 'Active' monsoonal conditions than observed from Amazon convection (e.g., May and Ballinger, 2007; Dolan et al., 2013; Schumacher et al., 2015; Giangrande et al., 2014a, 2016b). However, it appears use of BR would minimize the contributions from congestus or shallower organized convective precipitation found under Amazon conditions.

More recently, Thompson et al. (2015) highlighted limitations for BR concepts if characterizing oceanic precipitation observed over ARM Tropical Western Pacific (TWP) ground disdrometers at Manus island and Equatorial Indian Ocean Gan islands. Thompson et al. (2015), herein TM, proposed a unique oceanic convective-stratiform segregation having origins in LWC and $D_0$ space. One justification for this change was to better isolate DSD clusters exhibiting the higher concentrations of





smaller drops consistent with oceanic-convective clouds favoring warm-rain processes/collision-coalescence over mixed-phase and/or stratiform particle growth. The TM classification is simple to implement, since it overlaps within the BR space as a line of constant $\log10(N_w) \simeq 3.85\,\mathrm{m}^{-3}\mathrm{mm}^{-1}$. Figure 8 orients this dataset in BR/TM formulation spaces to the left of the BR line, with the DSDs identified as belonging to convective or stratiform (based on the RWP definitions) noted on the panels. When

populations from the Amazon DSDs exhibit more oceanic qualities (residing above the dashed TM line), contributions to the histograms (Figures 8a, 8c) are typically associated with RWP convection signatures. Similarly, DSDs identified as stratiform by the RWP (Figures 8b, 8d) follow those residing below the TM criteria for oceanic-like stratiform precipitation. Overall, bulk Amazon precipitation carries several hybrid characteristics as found from previous ARM tropical DSD studies.

### 4.2 Cumulative Precipitation Properties According to Cloud Regime and Season

Extending the previous analysis into cloud regimes, Figure 9 separates Amazon precipitation according to ETH values above/below 9 km. This choice follows the discussion from Figure 5 and is assumed as a reasonable proxy to also help separate statistical congestus from deeper convective events. These plots include combined convective precipitation (e.g., stronger updraft/downdraft regions) as well as associated trailing stratiform DSDs and/or decaying convection.

Figure 9 indicates that deeper cumulus clouds are associated with additional maritime continental DSD properties as similar to Darwin studies, with fewer observations residing above TM recommendations for possible oceanic characteristics. Deeper

convective and stratiform DSDs as designated by the RWP exhibit more frequent DSD examples having larger median drop sizes. In contrast, DSDs associated with ETH < 9 km carry DSD properties most similar to TM oceanic characteristics, having corresponding stratiform DSDs (or, the absence thereof) that also favor smaller median drop sizing than deeper column counterparts. While tempting to attribute these oceanic ETH < 9 km DSD characteristics solely to weak, isolated congestus

clouds, inspection of the events reveals oceanic DSDs are often associated with widespread convective lines and/or widespread convective cells (to be further discussed).

Figure 10 illustrates this cloud segregation according to Dry, Wet and adjacent months (here, 'Transitional' implying May, October and November properties that share qualities of both Wet/Dry seasons). The Dry season conditions (Figures 10a, 10d) skew towards bulk precipitation properties associated with the deeper convective clouds from above. These properties follow

an isolated, stronger convective cell expectation for Dry season precipitation, that also includes an absence of DSDs associated with detrained stratiform precipitation processes (e.g., low $N_w$, larger $D_0$) as to be discussed in the following section. In contrast, Wet season DSD characteristics (Figures 10b, 10e) follow previous tropical and oceanic expectations, with additional excursions into DSD contributions associated with the convective core modes (right of BR).

### 4.3 Stratiform Precipitation Properties Associated with Amazon Convective Events

Stratiform precipitation within the Amazon is commonly observed during the Wet season and adjacent months, associated with the detrained regions from deeper convective cells or cell dissipation. Increased stratiform precipitation frequency during the Wet season is attributed to factors including the seasonal change in midlevel moisture and reductions in Wet season convective inhibition more supportive of convective initiation and prevalence. Recalling Figures 8b and 8d, stratiform DSDs as identified





by the RWP often are the same as combining thoughts from BR/TM recommendations. This statement is further confirmed consulting cumulative and fractional convective precipitation as in Table 3. Figure 11 presents the composite DSD properties as reported in Figure 9, exclusive to RWP-indicated stratiform properties. Contours overlaid on Figure 11 indicate those DSD regions designated as having a bright-band signatures in the column. As from the left panels in Figure 11 (ETH > 9 km),

locations with profiles exhibiting clear bright band signatures correspond well with BR expectations for stratiform precipitation; For example, these would often represent the DSDs within more developed precipitation trailing deeper convective cells, mesoscale convective systems (e.g., Houze, 1997).

Lower echo-top stratiform characteristics (ETH < 9 km) indicate two unique clusters. The first cluster represents observations associated with aggregation processes that produce stronger melting layer signals, similar to ETH > 9 km examples.

These observations are found under Wet season conditions (50 % of the available DSDs), and are less common under Dry season conditions (30 % of the available DSDs). Initially, this supports an argument that enhanced Wet season moisture influences sustained stratiform development, ice growth (deposition), and eventual aggregation processes. The second cluster is associated with smaller median drop sizes and higher-relative number concentrations. This represents the more prevalent stratiform mode for lower-top Dry season observations, and is equally frequent for Wet season observations. This cluster argues for less

developed stratiform processes, either owing to the lack of mid-level moisture in Dry season profiles, or consistent with Wet season widespread, weaker congestus (e.g., reduced inhibition resulting in larger areas having weaker updraft intensity).

### 4.4   Implications of Convective-Stratiform Partitioning

Previous sections indicate that RWP and hybrid BR/TM classifications are those that faithfully differentiate congestus and deeper convective DSDs from stratiform DSDs. Table 3 reports the total convective precipitation and fractional convective

precipitation for this GoAmazon2014/5 dataset. These values are estimated according to segregations from BR methods, a hybrid BR/TM combination, the RWP classification, and a simple rainfall rate $R > 10$ mmhr$^{-1}$ threshold. Table 3 has also been segregated according to Wet/Dry and Transitional season component behaviors.

For the Amazon dataset, both TM/BR and RWP methods attribute approximately half of the total precipitation (convective plus stratiform) to possible congestus or shallower cloud regimes, as defined by our ETH < 9 km definitions. Moreover, we ob-

serve that the fractional convective precipitation is higher for those methods adding additional complexity to the classification. Convective fractions suggest differences to within $\simeq 10$ %. Seasonal breakdowns confirm that the Wet season and adjacent months are more dominated by stratiform rainfall, with transitional months suggesting the largest share of stratiform precipitation. Overall, fractional convective contributions are high (exceeding 80%), but the strong agreement between RWP and BR/TM ideas gives confidence that traditional radar segregations would report lower convective fractions owing to incorrect

attribution of congestus or shallower-topped precipitation systems.

It is possible to check whether dual-polarization radar quantities are sensitive to apparent variations between congestus, deeper convection and associated stratiform precipitation properties. Figure 12 plots ($Z$, $Z_{DR}$) scatterplot as well as a ($K_{DP}$-$Z$-$Z_{DR}$) self-consistency curve behaviors for various regimes identified by the RWP; Lower panels in Figure 12 illustrate the Wet and Dry season segregations. For all panels in Figure 12, we present X-band dual-polarization estimates calculated from



T-Matrix scattering, as radar quantities at these shorter wavelengths should be more sensitive to lower rainfall rate conditions. A more practical consideration for these ideas is to support future studies from a dual-polarization X-band radar that was operated during GoAmazon2014/5. The radar quantities are presented in terms of their associated two-sigma confidence regions (ellipses). Since radars routinely perform separate ETH and/or bright-band designation checks, the demonstrations in Figure 12 are not a true reference for what is possible from a robust radar echo classification methodology. However, Figure 12 suggests substantial overlap between these cloud precipitation regimes when placed in this dual-polarization context. This would suggest X-band or longer-wavelength radars would not be sufficient constraints for regime classifications without additional information. The most pronounced contrasts are those observed between Wet/Dry seasons, wherein the Dry season favors the larger extremes for all dual-polarization radar quantities, associated with the contributions of larger drops.

## 5 Amazon Precipitation Properties: The Green Ocean Characteristics

The Amazon Wet season has been highlighted for its copious precipitation owing to factors including enhanced moisture and reduced convective inhibition (CIN). One additional consideration is that these conditions, possibly when coupled with cleaner atmospheric aerosol profiles, may promote the so-called 'Green Ocean' or oceanic cloud and precipitation characteristics. In contrast, Dry season convective conditions migrate towards enhanced Convective Available Potential Energy (CAPE) and stronger CIN that may promote stronger convective storms, initiating within more polluted atmospheric states closer to continental regimes. Other recent Amazon studies that indicate that the convection that initiates during the Amazon Dry season exhibits more intense vertical air motions and precipitation properties (e.g., Giangrande et al., 2016b; Schiro, 2017).

### 5.1 The Amazon 'Green Ocean': When Do We Observe Oceanic Behaviors?

Figure 13 extends the previous analysis found in Figure 10 to a seasonal comparison between deeper clouds (ETH > 9 km, reds) and congestus or shallower convection (ETH < 9 km, blues). To simplify, 'stratiform' DSD components (as identified by the RWP) have been removed from this figure. Although all DSDs are assumed as 'convective', it is instructive to focus on DSDs in Figure 13 located to the right of the BR separation line, as those DSDs correspond to the most confident convective conditions having typical rainfall rate $R > 13 \ \mathrm{mmhr}^{-1}$. As also in Table 1, convective Dry season DSDs carry fewer drops, but larger median drop sizes. Physically, this corresponds well with expectations that stronger updrafts in the Dry season should promote larger droplet sizes as a consequence of mixed-phase growth. Wet season characteristics are noticeably shifted towards higher number concentrations, with lower-relative LWC. This is consistent with the anticipated changes towards more oceanic and/or tropical warm-rain processes, cleaner and/or weaker updraft storms. For dual-polarization radar studies, these characteristics are consistent with Dry season convection exhibiting larger values in $Z_{DR}$ or $K_{DP}$ for a similar value of $Z$, noting surface conditions may also be modified slightly from the conditions sampled aloft from radar.

Figure 14 plots congestus and deep convective full DSD averages for convective conditions as in Figure 13. Average DSDs are also provided for those observations found to the right of the BR separation line, as well as those DSDs having $Z > 35$ dBZ. Overall, composite behaviors emphasize that Dry season convective precipitation (and into convective core regions) is



skewed towards an increased presence of larger drops, parameter spaces favoring higher LWC for a similar $D_0$. In contrast to Wet season properties, Amazon Dry season precipitation conditions are not consistent with TM oceanic findings (shift towards DSDs having increased larger drops), though do support that the updrafts in the Dry season are stronger.

### 5.2 The Amazon 'Green Ocean': Role of Pollution on Oceanic Signatures?

Overall, the primary shift in precipitation properties for the Amazon coincides with changes in the larger-scale seasonal shifts in thermodynamics and aerosol conditions. In this respect, it is difficult to differentiate relative controls, especially given sampling limits of our Amazon precipitation dataset during the Dry season. However, the frequent Wet season convective instances (removing the more obvious stratiform contributions) offers some opportunity to test whether we observe any sensitivity to background aerosol conditions and/or other environmental conditions when promoting so-called oceanic DSD properties.

Figure 15 plots the set of convective DSDs observed during the Wet season, identifying the relative clean (blues) and polluted (reds) aerosol conditions. Panels beneath the cumulative convective plots illustrate the convective DSDs associated with column ETH < 9 km. Rightmost panels on Figure 15 plot a composite median, 90th and 95th percentile RWP $Z$ profile under the clean and polluted conditions, respectively. For simplicity, polluted regimes in our study combine the more stringent (but, in this dataset, the more frequent) 'biomass' polluted classification with standard 'polluted' designations. During this campaign, a

total of 82 clean and 61 polluted events were collected having at least one 5-minute convective DSD, with 66 clean events registering an ETH < 9 km DSD and 46 polluted events with a ETH < 9 km DSD, respectively.

The mean thermodynamic conditions are sampled from the morning 12 UTC radiosondes. For this dataset, clean events record a mean (standard deviation) most unstable MUCAPE of 2124 (1100) $\mathrm{Jkg^{-1}K^{-1}}$, most unstable MUCIN of -34 (42) $\mathrm{Jkg^{-1}K^{-1}}$, and average 0-5 km RH of 83 (6) %. Polluted events are slightly more favorable to deeper convection, in recording

a higher mean MUCAPE of 2567 (1176) $\mathrm{Jkg^{-1}K^{-1}}$, with an MUCIN of -35 (36) $\mathrm{Jkg^{-1}K^{-1}}$ and RH of 80 (7) %, respectively. Both clean and polluted events share a similar mean freezing level height at approximately 4.8 km. For the ETH < 9 km panels, mean clean (polluted) environments appear less favorable, with MUCAPE of 1993 (2388) $\mathrm{Jkg^{-1}K^{-1}}$, MUCIN of -36 (-38) $\mathrm{Jkg^{-1}K^{-1}}$, and RH of 83 (81) %. Standard deviations for clean (polluted) values are similar as ETH > 9 km convection.

Figure 15 indicates that cleaner regime convective precipitation during the Wet season agrees well with oceanic expecta-

25 tions as reported by TM and discussions above. Cumulative polluted regime convective results are less consistent with oceanic expectations, but there is overlap emphasizing DSDs associated with ETH < 9 km columns. Deeper ETH > 9 km polluted convective observations (deeper convection properties) are those most skewed towards Dry season and/or least oceanic behaviors, including hints of stratiform-type DSD excursions. Inevitably, some DSD contamination could follow from convective-to-stratiform transitional columns in the strongest storms as well, for example those featuring sloped updrafts having stronger

vertical motions aloft overhanging a stratiform-type downdraft in the column below

Bulk clean/polluted contrasts are potentially visible on the composite $Z$ profiles, with cleaner regime composites demonstrating an increasing $Z$ profile ($Z$ weighted towards increasing contributions from larger drops) towards the surface. One explanation is that these cleaner profiles more routinely are associated with collisional growth process contributions influencing $Z$ profiles over evaporation and/or breakup process influences on radar signatures (e.g., evaporation and/or breakup acting





to reduce $Z$, perhaps not observable with available larger drops to RWP wavelengths). These profile behaviors are pronounced for the ETH < 9 km observations that should minimize mixed-phase process influences. In contrast, the polluted regime profiles indicate similar and/or larger $Z$ values aloft to approximately 3.0 km AGL, with $Z$ profiles peaking and/or decrease in magnitude below these altitudes.

5     One explanation for the polluted profile characteristics in Figure 15 are more prominent mixed-phase particle process contributions acting within these convective columns. Since these polluted events demonstrate more favorable mean thermodynamic conditions that favor stronger convective updrafts, it is possible that an updraft enhancement partially elicits such a transition. A similar response may also be attributed to the proposed role of aerosols in following invigoration arguments (e.g., Rosenfeld et al., 2008). For example, recent Amazon aircraft studies as in Braga et al. (2017) indicate changes such as an absence of liquid within growing convective cumulus during polluted conditions, and/or differences in the relative formation/altitudes of ice particles. Regardless of process path, the suggestion is that polluted convective columns would be those that potentially promote added ice depositional growth (resulting in fewer, but larger ice particles at the expense of additional liquid). Such physical arguments could help explain the similar or larger $Z$ magnitude aloft (larger ice sizing, offsetting density), coupled with a modest melting enhancement followed by a reduction in $Z$ below 5 km. A reduced number of particles under this scenario would also reduce collisional growth below the freezing level as compared to the cleaner profiles. Overall, surface DSD properties in Figure 15 suggest cleaner-aerosol conditions as associated with enhanced oceanic DSD properties (aka, in agreement with select 'Green Ocean' statements). However, it is nonobvious whether these lesser oceanic conditions (esp. within the deeper cores having fewer samples) were the consequence of the aerosol conditions, or the shift in the environmental conditions that tracked the change in aerosol.

20 **5.3   The Amazon 'Green Ocean': An Alternate Explanation**

It is useful to determine whether we can better deconvolve environmental influences from aerosol as those more important to the prevalence of oceanic precipitation characteristics. Figure 16 plots Wet season DSDs contingent on the ambient wind directions, with relative breakdowns according to the northeasterly/east-southeasterly (NE/ESE) and east/east-northeast (E/ENE) directional pairings. First, the specific (NE/ESE) and (E/ENE) pairings were selected for having similar DSD sample sizes. Second, these wind orientations may also be viewed as relevant with respect to the Manaus pollution plume (e.g., E and ENE flows over T3 as arguably the more polluted relative to the Manaus location).

Figure 16 highlights evidence of oceanic-type DSD behaviors according to most wind directions. The fractional 'polluted' versus 'clean' DSD breakdowns along these directions are as follows: NE: 57 % clean 43 % polluted; ENE: 68 % clean 32 % polluted; E: 94 % clean 6 % polluted; ESE: 91 % clean 9 % polluted. Following Figure 16, it is found that the larger DSD outlier populations (e.g., convective DSDs found to be least 'oceanic' when compared with TM) are observed for NE and ESE wind directions that should not be as influenced by possible Manaus pollution plume. Note, most polluted events sampled during the Wet season were attributed to 'biomass' classifications, e.g., local aerosol sources, which may explain NE flows as those most polluted. As expected from discussions above, slightly stronger 12 UTC MUCAPE (STD) values are also found along the NE and ENE directions (2207 (1325) $\mathrm{Jkg^{-1}K^{-1}}$ and 2131 (934) $\mathrm{Jkg^{-1}K^{-1}}$ , respectively) that are associated with bulk polluted





events, while the weakest potential forcing conditions are found with the ESE and E flows (2089 (1241) $\mathrm{Jkg^{-1}K^{-1}}$ and 1766 (1035) $\mathrm{Jkg^{-1}K^{-1}}$, respectively). Nevertheless, these local thermodynamic controls associated with wind direction are far less pronounced than previous polluted/clean contrasts.

The DSDs observed along NE wind flows reflect the least oceanic characteristics in this dataset, favoring low $N_w$-$D_0$ pairings typical of Dry season convection (also carrying similar $Z$ profiles as to Figure 15, not shown). Again, these NE flows reflect the most polluted wind components, and directions associated with the larger convective forcing parameters. In that regard, a reduced presence for oceanic-type DSDs was not unexpected. However, the pronounced absence of oceanic DSD characteristics along NE flows is far more noteworthy than when contrasted to previous clean/polluted criteria, and not in line with local thermodynamic changes. From event inspection, most nonoceanic DSD characteristics were associated with isolated, deeper convective cell events, or widespread convective events still demonstrating deeper cloud ETH. Widespread, shallower convective events or organized shallow systems (possible Amazon warm-rain dominant systems as observed over oceans, e.g., Chen and Liu (2015)) were not favored, as compared with other wind components.

Additional outlier DSD populations (including several events having numerous oceanic DSD properties) are observed according to ESE wind directions (relatively clean). These DSDs reflect the presence of deeper convective DSDs (to the right of the BR separation line) that exhibit high concentrations of larger-relative drop sizes. These regions, although not typical of TM oceanic examples, are also not consistent with Amazon Dry season characteristics (having relatively higher triplet of LWC, $D_0$ and $N_w$). As in NE flow examples, the basic radiosonde parameter checks and aerosol forcing controls associated with these events are in-line with the other wind components.

As far as potential explanations for these outliers to cluster according to particular wind directions as compared to other environmental factors, it is important to note that while Amazon convection timing follows a well-established diurnal cycle over T3, 12 UTC radiosondes and associated parameters (those typically closest to earlier convective initiation) may not be completely representative of the important larger-scale conditions (e.g., South Atlantic Convergence Zone (SACZ) positioning, influences into the Amazon basin during the Wet season, Carvalho et al. (2004)). For one example, Wet season sea-breeze intrusion and associated statistical cloud enhancements (as determined by satellite) into the Amazon basin orient tangential to a NE-SW axis over the T3. This sea-breeze front passage is in phase with this T3 diurnal precipitation cycle (e.g., see composite convective evolution as in Burleyson et al. (2016)). It is possible that similar forms of dynamical or moisture enhancements, for example SACZ drivers of frontal intrusions, as well as river breeze influences (e.g., Tanaka et al., 2014; Burleyson et al., 2016), would not be completely captured by our morning radiosonde observations (given their timing). However, these larger-scale features may promote enhancements sufficient to spark possible changes in cloud initiation, subsequent precipitation properties.

From inspection of events according to wind directions, ESE events tended to emphasize widespread organized convective events exhibiting copious rainfall along a NE-SW orientation (with winds flowing from ESE preceding those lines), having shallower ETH < 9 km. Timing for these events were near or just following the afternoon diurnal maximum (18-20 UTC). One suggestion is that the oceanic DSDs tended to be those associated with these shallower, but widespread convective events initiated or enhanced by sea-breeze influences. As the conditions are also clean, this is also consistent with shallower, oceanic





forms of organized convection. These combined concepts and possible SACZ influences on these events are the subject of ongoing research. In contrast, NE events most often reflected deeper events (ETH > 9 km) with less evidence for forms of NE-SW linear or shallow cloud organization for animations of the widespread events. Deeper clouds would be consistent with pollution arguments as above, but these breakdowns speak to the complexities of these studies.

5 **6 Conclusions**

This study summarizes Amazon precipitation properties collected during the unique, multi-year GoAmazon2014/5 campaign. Emphasis was placed on cumulative campaign precipitation properties and relationships that may benefit potential hydrological applications and radar-based precipitation data product development, as well as connections relevant to future Amazon convective model evaluation. The study also explored Amazon precipitation properties from the perspective of possible 'Green 10 Ocean' convective characteristics, including possible thermodynamic and aerosol forcing influences that may be influential to observations of oceanic-like precipitation properties.

Amazon rainfall and radar self-consistency relationships demonstrate tropical characteristics as compared to continental SGP references, associated with radar quantities (in both convective and stratiform contexts) that sample higher relative concentrations of smaller drops. Typically, this indicates a reduced role for convective mixed-phase and/or graupel growth, as well 15 as stratiform aggregation processes in the Amazon. These tropical precipitation characteristics are more pronounced within the Wet seasons than Dry season events, with Dry season storms favoring the presence of larger drop sizes as a suggested consequence of stronger storm updrafts under more favorable thermodynamic conditions. Although it is difficult to differentiate Wet/Dry regimes exclusively using radar quantities, our analysis suggests $Z$, $Z_{DR}$ and $K_{DP}$ would exhibit larger values within Dry season events and deeper convective cores therein.

20 Coupled RWP and disdrometer Amazon T3 precipitation breakdowns confirm the overall findings of previous ARM campaign BR and TM studies on tropical convective to oceanic type cloud and precipitation breakdowns. Amazon precipitation is varied and often found to straddle maritime continental behaviors of previous studies, with DSD excursions into the more oceanic examples presented from ARM Manus/Gan deployments. As before, the separations between Wet and Dry seasons are pronounced, with most oceanic DSD conditions observed during the Wet season. The strongest convective behaviors, as well 25 as storms having a marked absence of stratiform precipitation, are observed during Amazon Dry season.

Consulting deeper versus congestus properties, Amazon congestus are attributed to the more oceanic precipitation behaviors found in our dataset. When exploring 'Green Ocean' themes, our analysis was not able to demonstrate that either aerosol conditions or enhanced local radiosonde convective forcing parameters were strongly associated with the presence/absence of an oceanic character to the congestus and deeper precipitation. Rather, the more pronounced separation was found when 30 segregating on wind direction, which may reflect that our initial options for thermodynamic or aerosol controls are all unable to deconvolve a more subtle change important to an enhanced DSD signature. However, there is evidence to support that aerosol or other early morning forcing factors within the Wet season are not significantly different to promote these differences. Rather, episodic to frequent Amazon basin larger-scale (e.g., SACZ, sea-breeze) or river forcing controls and associated enhancements




may require future investigation to determine their importance to the apparent oceanic nature of the clouds and eventual precipitation. Other factors including the possible role of aerosol sizing (e.g., Fan et al., 2018) on updraft and precipitation enhancements for Amazon convection are also the topic of future consideration.

*Data availability.* All ARM datasets used for this study can be downloaded at http://www.arm.gov and associated with several "value added

5  product" streams.

*Competing interests.* The authors declare that they have no conflict of interest.

*Acknowledgements.* This paper has been authored by employees of Brookhaven Science Associates, LLC, under contract no. DE-SC0012704 with the U.S. Department of Energy (DOE). The publisher by accepting the paper for publication acknowledges that the United States Government retains a non-exclusive, paid-up, irrevocable, world-wide license to publish or reproduce the published form of this paper, or allow

10  others to do so, for United States Government purposes. Dr. Joseph Hardin and Dr. Zhe Feng at the Pacific Northwest National Laboratory (PNNL) are supported by the Climate Model Development and Validation activity funded by the Office of Biological and Environmental Research in the U.S. Department of Energy Office of Science also acknowledge the Atmospheric Radiation Measurement (ARM) Climate Research Facility, a user facility of the U.S. DOE, Office of Science, sponsored by the Office of Biological and Environmental Research, and support from the ASR program of that office.





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





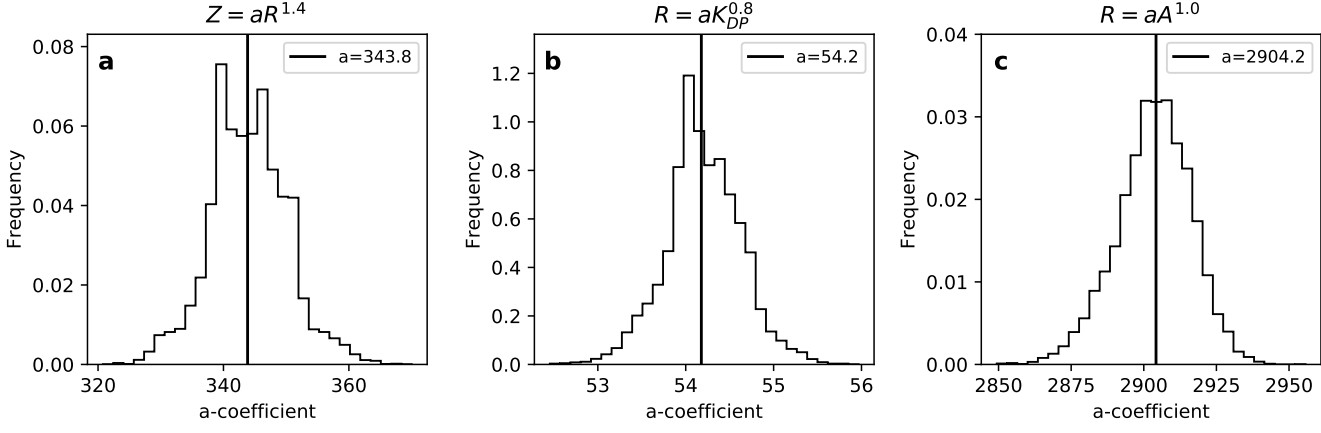

**Figure 1.** Histograms for a-coefficient values from single parameter rainfall relationships (a) $R(Z)$, (b) $R(K_{DP})$, and (c) $R(A)$, calculated using least square method under the assumption of a fixed b-coefficient from random sampling of half of the dataset (5000 times), for the S-band wavelength. The black vertical lines represent the a-coefficient calculated based on the whole dataset.





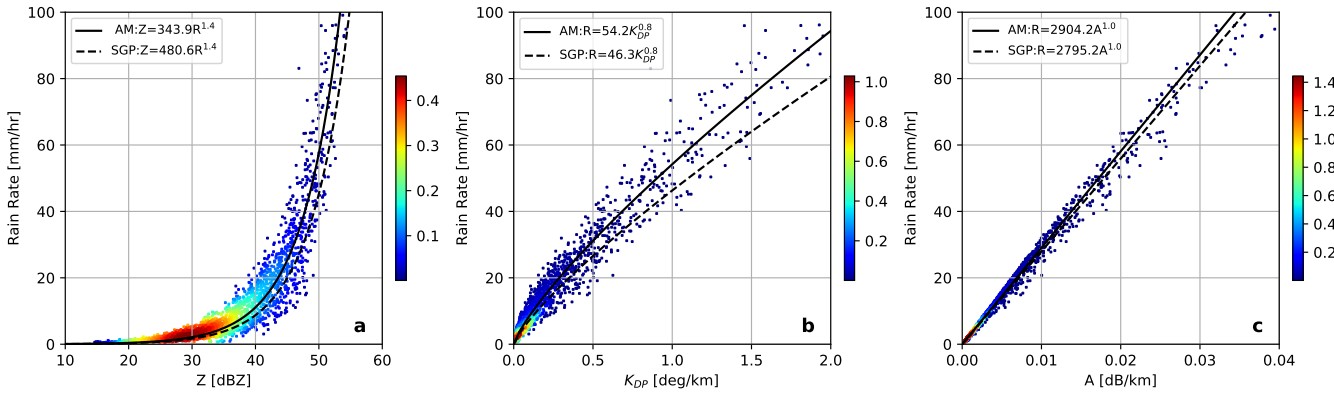

**Figure 2.** Scatter plots of (a) $Z$, (b) $K_{DP}$, and (c) $A$ versus rain rate and overlaid associated relationship fits using least square method for Amazon (AM, solid lines) and SGP-Oklahoma (SGP, dashed lines) sites, for the S-band wavelength. Density calculated using a kernel function is shown in color.



**Figure 3.** Contoured frequency altitude display histograms (CFADs) for the entire Amazon dataset with confidence intervals, median (thick black lines), 90th (thin black lines) and 95th percentile (white lines), for RWP convective and stratiform reflectivity profiles (a, b) and vertical velocity retrievals (c, d). The number of profiles of each situation is shown as a red line.



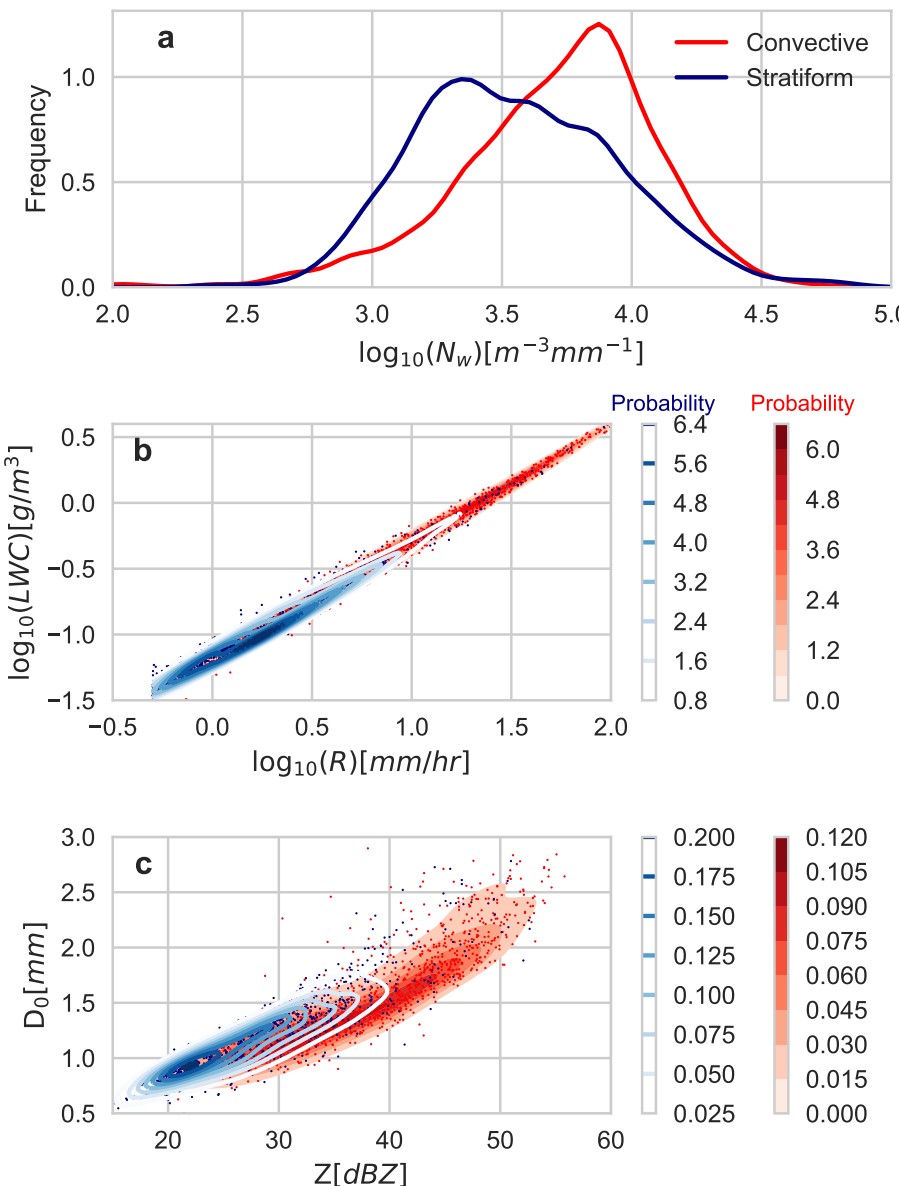

**Figure 4.** Histograms associated with RWP classification based convective (red) and stratiform (blue) DSDs in terms of $N_w$ (a), LWC versus $R$ behaviors (b), and $D_0$ scaling according to $Z$ (c).



**Figure 5.** Histograms associated with RWP classification based convective DSDs in terms of ETH (a), temperature at ETH (b), and *Z* at 2 km (d) for All, Dry, Wet, Transitional seasons, as well as the congestus for all the seasons. The wind rose (c) is also shown for all the seasons.





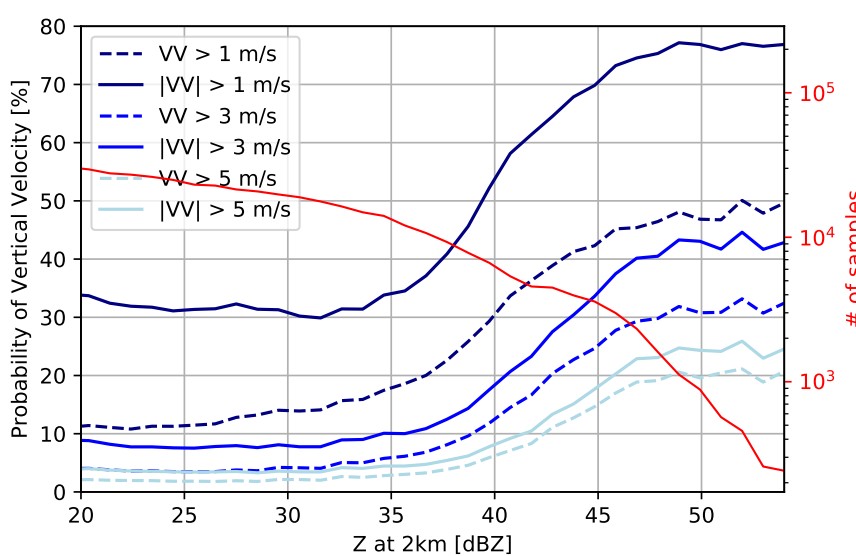

**Figure 6.** The frequency for observing a given vertical velocity (across all levels, $> 1 \ \mathrm{ms}^{-1}$ in navy, $> 3 \ \mathrm{ms}^{-1}$ in dark green, $> 5 \ \mathrm{ms}^{-1}$ in light green) as a function of a 2 km RWP reflectivity. The number of samples (for $|VV| > 1 \ \mathrm{ms}^{-1}$) are displayed as a red line.



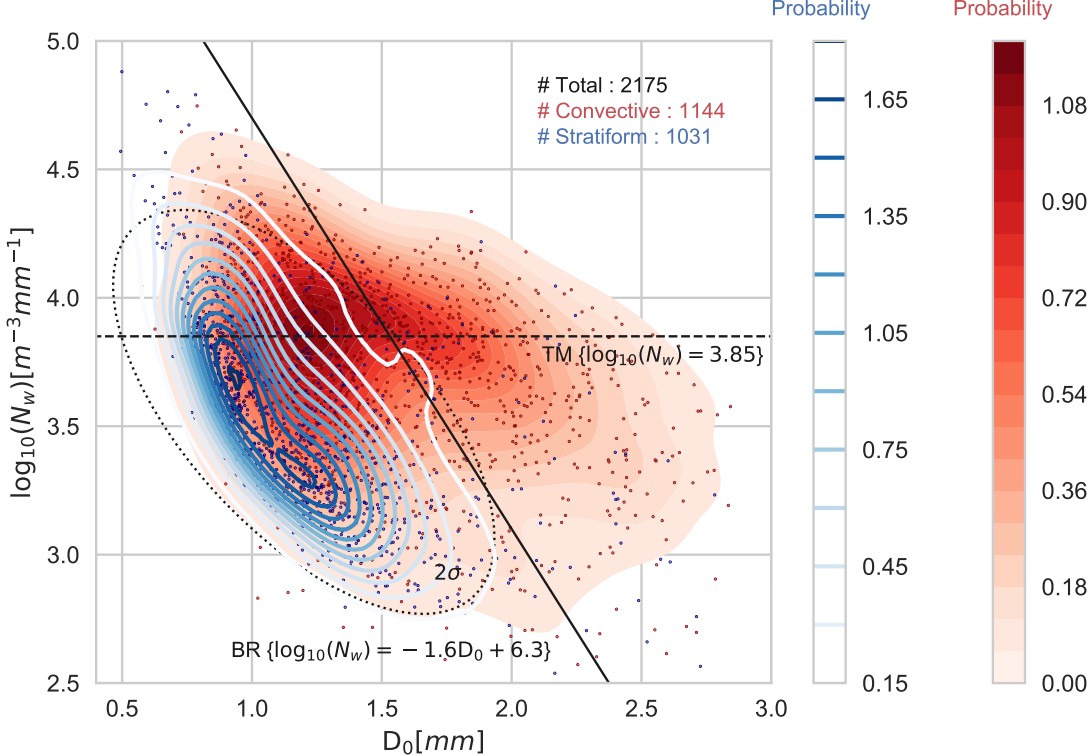

**Figure 7.** Scatter plot of $\log_{10}(N_w)$ versus $D_0$ from PARSIVEL disdrometer, overlaid by the contours representing the RWP-based classifications for convective (red colors) and stratiform (blue lines) precipitating columns. The ellipse conveys the two-sigma confidence interval (dotted line) for those regions containing RWP-based stratiform DSDs. The convective-stratiform regime segregation concepts in Bringi et al. (2003, BR) and in Thompson et al. (2015, TM) are presented as a solid black line and dashed black line, respectively.







**Figure 8.** Scatter plots of $\log_{10}(N_w)$ versus $D_0$ and LWC versus $D_0$ for BR-based stratiform DSDs (probability in colors). DSDs identified as convective/stratiform by the RWP are shown in (a, c)/(b, d). BR line and TM line are shown as a solid black line and dashed black line, respectively.





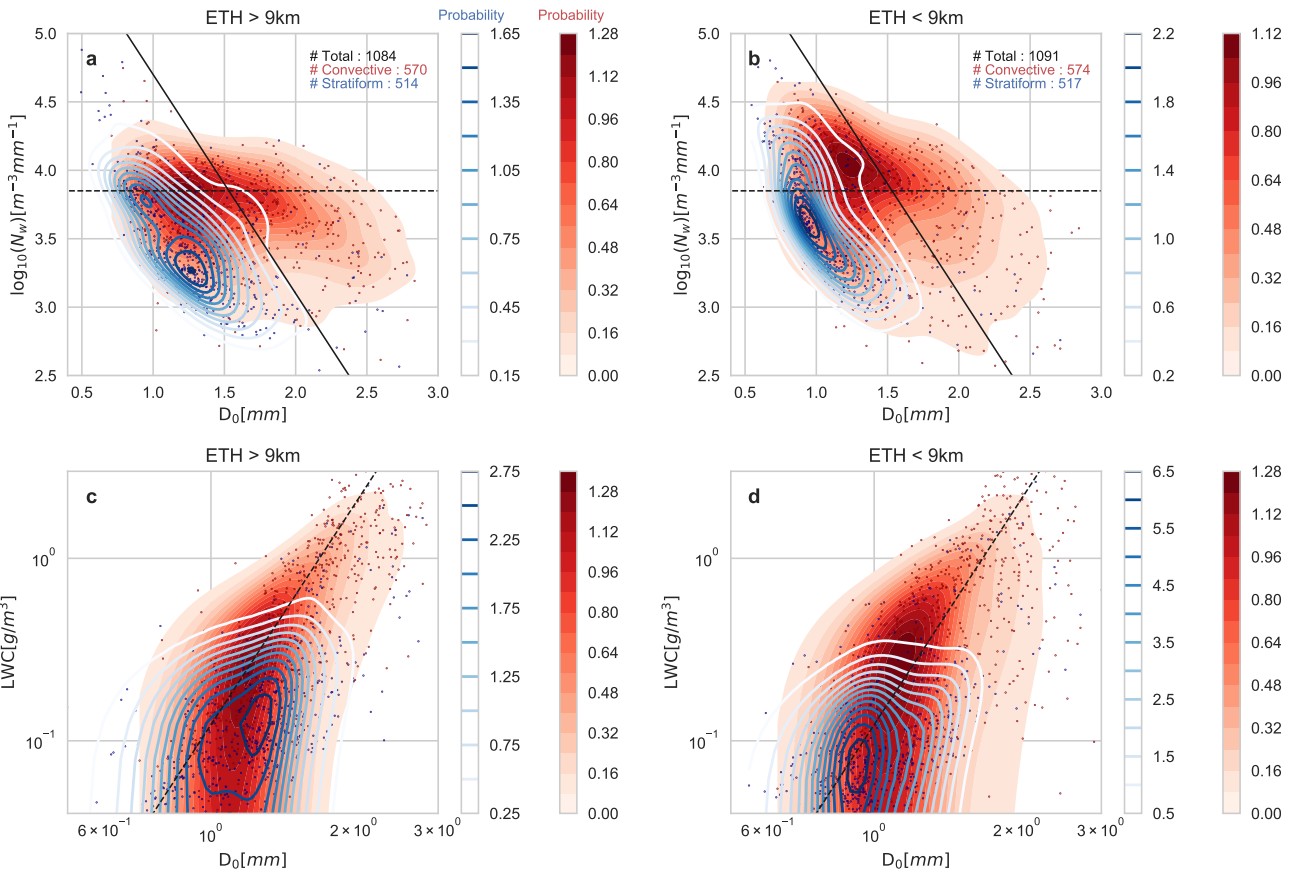

**Figure 9.** Scatter plots of $\log_{10}(N_w)$ versus $D_0$ and LWC versus $D_0$, overlaid by the contours representing the RWP-based classifications for convective (red colors) and stratiform (blue lines) precipitating columns, for ETH > 9 km (a, c) and ETH < 9 km (b, d) situations. BR line and TM line are shown as a solid black line and dashed black line, respectively.





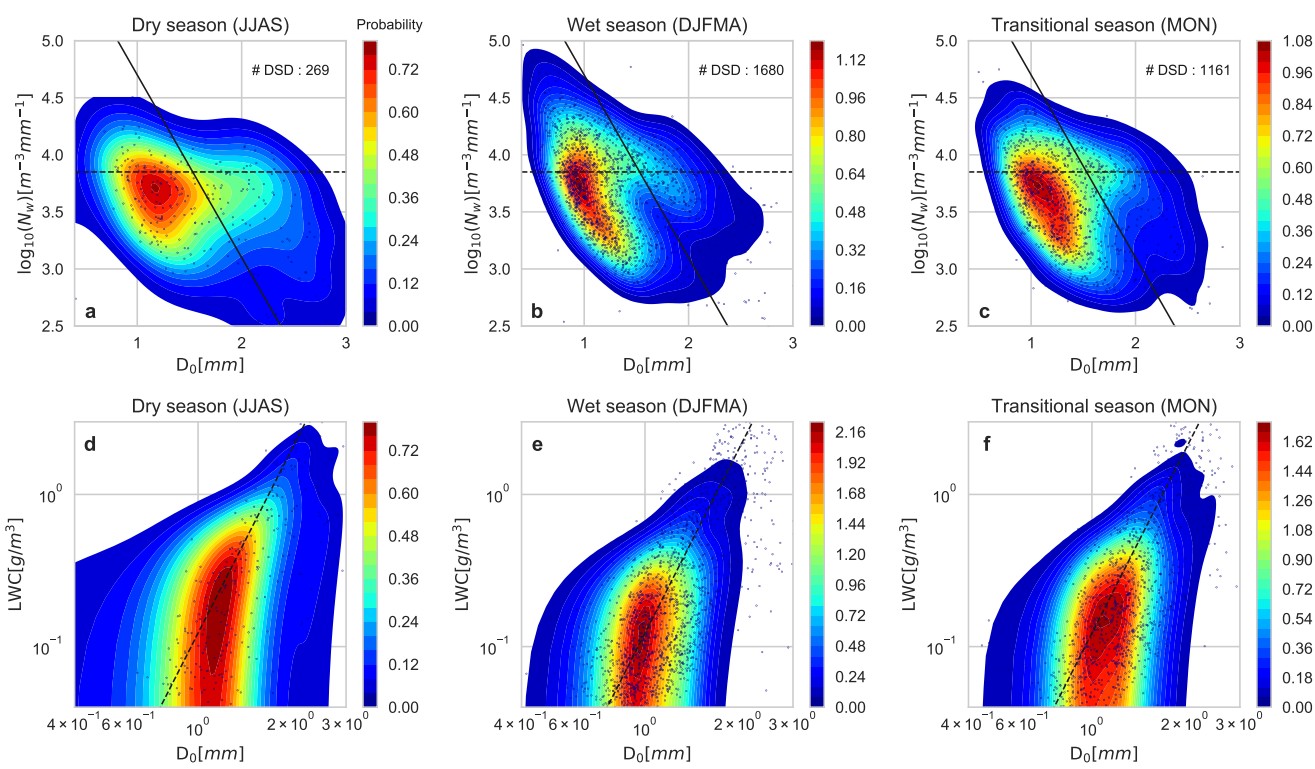

**Figure 10.** Scatter plots of $\log_{10}(N_w)$ versus $D_0$ and LWC versus $D_0$ for Dry (a, d), Wet (b, e), and Transitional (c, f) seasons (probability in colors). BR line and TM line are shown as a solid black line and dashed black line, respectively.



**Figure 11.** Scatterplots of $\log_{10}(N_w)$ versus $D_0$ and LWC versus $D_0$ for RWP-based stratiform DSDs (probability in colors), for ETH < 9 km (a, c) and ETH > 9 km (b, d) DSDs. The overlaid black contours represent the RWP-based classifications for stratiform with bright band precipitating columns. BR line and TM line are shown as a solid black line and dashed black line, respectively.





**Figure 12.** Scatter plots of $Z_{DR}$ versus $10\log_{10}(K_{DP}/Z)$ and $Z$ versus $Z_{DR}$ for the various regimes, deep convection, congestus, stratiform with bright band and stratiform without bright band identified by the RWP classifications (a, b). The ellipses convey the two-sigma confidence interval for corresponding regimes. The Wet (shaded ellipses)/Dry (ellipses) season segregations (c, d) are presented in c and d.





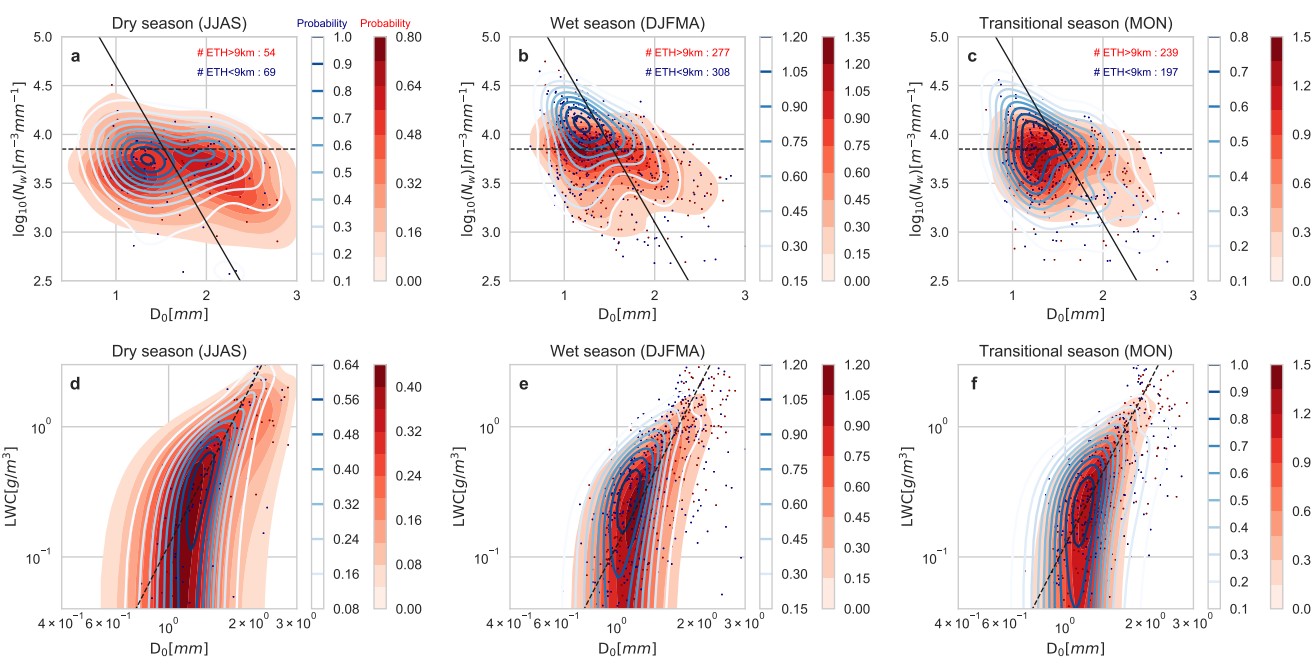

**Figure 13.** Scatter plots of $\log_{10}(N_w)$ versus $D_0$ and LWC versus $D_0$ for RWP-based convective DSDs, for Dry (a, d), Wet (b, e) , and Transitional (c, f) seasons. The contours represent the congestus (ETH < 9 km, blues) and deep (ETH > 9 km, reds) convective DSDs. BR line and TM line are shown as a solid black line and dashed black line, respectively.



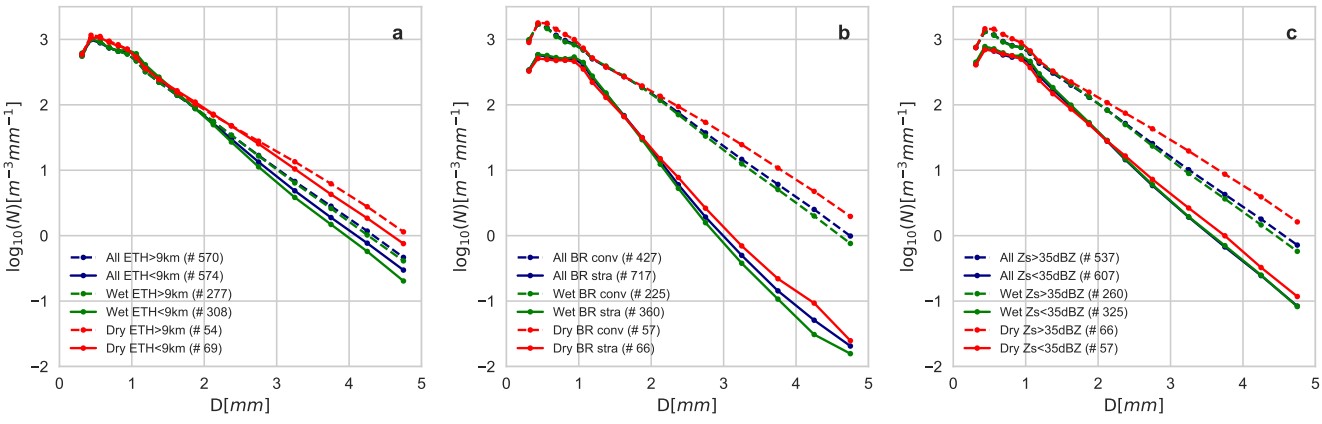

**Figure 14.** Averaged RWP-based convective DSDs for congestus (ETH < 9 km) and deep (ETH > 9 km) DSDs (a), for convective and stratiform DSDs depending on the BR separation (b), as well as for those having $Z$ (at surface) < 35 dBZ and $Z$ > 35 dBZ (c), for All, Dry and Wet seasons.





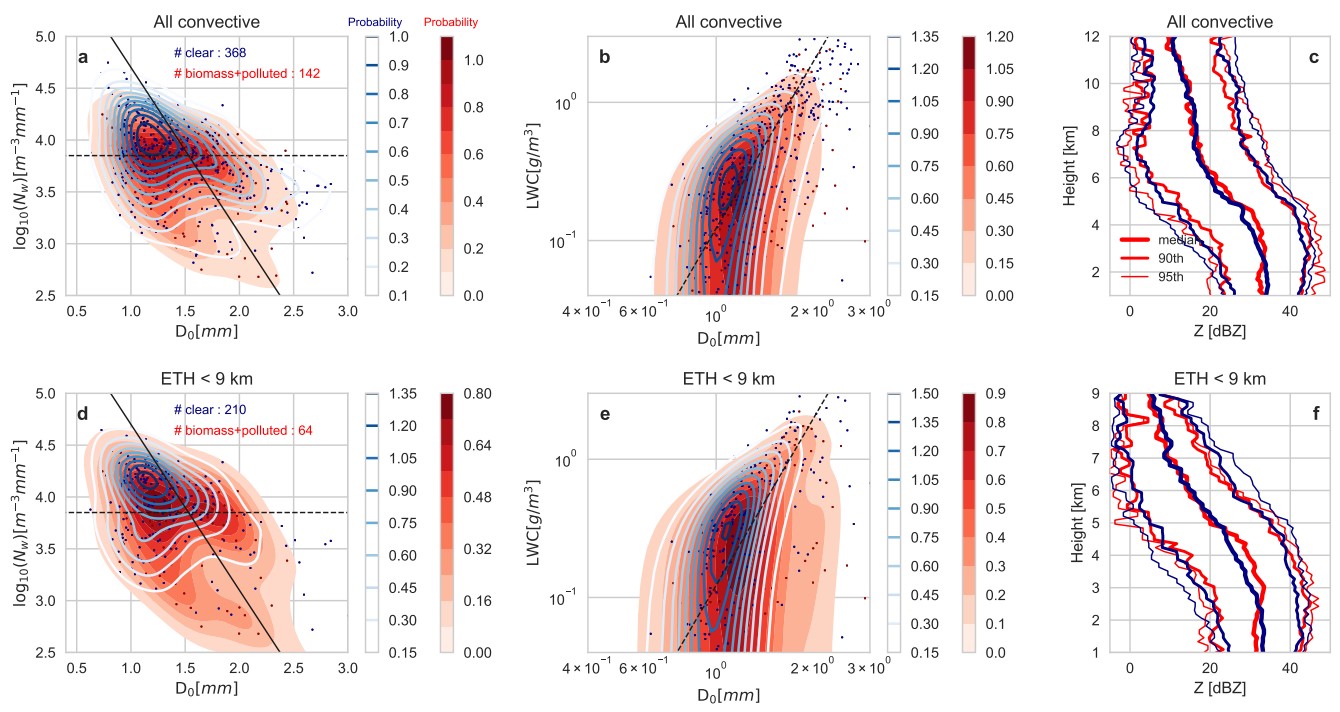

**Figure 15.** Scatter plots of $\log_{10}(N_w)$ versus $D_0$ and LWC versus $D_0$ for RWP-based convective DSDs under the Wet season (a, b), as well as for only congestus convective DSDs (d, e), contouring the clean (blues) and polluted (reds) conditions. The corresponding composite median, 90th and 95th percentile RWP Z profile behaviors under the clean (blue) and polluted (red) conditions are shown in c and f.





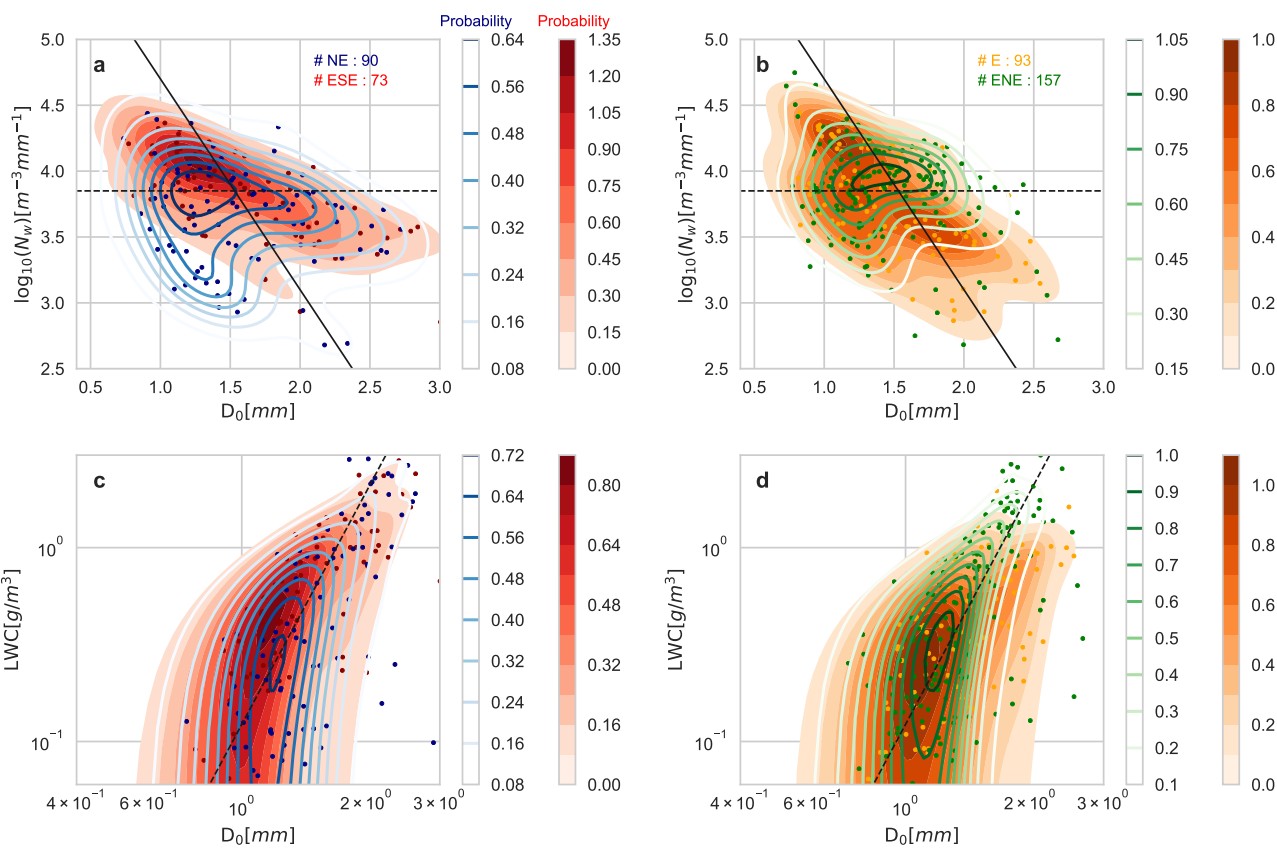

**Figure 16.** Scatter plots of $\log_{10}(N_w)$ versus $D_0$ and LWC versus $D_0$ for Wet season DSDs on the ambient wind directions, northeasterly/east-southeasterly (NE (blues)/ESE (reds)) and east/east-northeast (E (oranges)/ENE (greens)).



**Table 1.** A summary of 5-minute DSD parameter breakdowns for number of DSDs, rain rate $R$, median volume drop size $D_0$, normalized DSD intercept parameter $N_w$, reflectivity $Z$ at S-band, and liquid water content LWC, filtered according to rainfall rate intervals, for All, Wet, and Dry seasons.

| $R[mmhr^{-1}]$ | No. DSD | | $<R>[mmhr^{-1}]$ | | $<D_0>[mm]$ | | $<N_w>[m^3mm^{-1}]$ | | $<Z>[dBZ]$ | | $<LWC>[gm^{-3}]$ | |
|---|---|---|---|---|---|---|---|---|---|---|---|---|
| **All** (Total rainfall = 2597 mm for Amazon Dataset, 694 mm for SGP Dataset) | | | | | | | | | | | | |
| | AM | SGP | AM | SGP | AM | SGP | AM | SGP | AM | SGP | AM | SGP |
| 0.5-2 | 1080 | 676 | 1.15 | 1.17 | 1.01 | 0.97 | 6580 | 8882 | 24.1 | 24.1 | 0.08 | 0.08 |
| 2-4 | 582 | 337 | 2.86 | 2.87 | 1.24 | 1.26 | 6525 | 4718 | 30.5 | 30.9 | 0.17 | 0.15 |
| 4-6 | 294 | 148 | 4.83 | 4.83 | 1.34 | 1.46 | 7621 | 4454 | 33.7 | 34.7 | 0.27 | 0.23 |
| 6-10 | 292 | 116 | 7.66 | 7.56 | 1.49 | 1.73 | 7445 | 3873 | 36.5 | 38.5 | 0.39 | 0.34 |
| 10-20 | 339 | 85 | 14.61 | 14.29 | 1.70 | 1.95 | 6913 | 4333 | 40.8 | 42.3 | 0.69 | 0.61 |
| 20-40 | 289 | 61 | 27.79 | 28.48 | 1.90 | 2.16 | 6948 | 4543 | 44.9 | 45.9 | 1.21 | 1.08 |
| 40-60 | 93 | 19 | 48.95 | 47.89 | 2.07 | 2.24 | 7699 | 5502 | 48.4 | 49.0 | 2.03 | 1.82 |
| **Wet season** (Total rainfall = 1245 mm for Amazon Dataset) | | | | | | | | | | | | |
| 0.5-2 | 649 | | 1.14 | | 0.99 | | 6892 | | 23.7 | | 0.08 | |
| 2-4 | 301 | | 2.88 | | 1.19 | | 7851 | | 30.1 | | 0.17 | |
| 4-6 | 148 | | 4.80 | | 1.33 | | 8933 | | 33.5 | | 0.27 | |
| 6-10 | 147 | | 7.73 | | 1.49 | | 8295 | | 36.1 | | 0.39 | |
| 10-20 | 162 | | 14.78 | | 1.65 | | 8149 | | 40.4 | | 0.72 | |
| 20-40 | 147 | | 27.57 | | 1.86 | | 7666 | | 44.7 | | 1.23 | |
| 40-60 | 44 | | 49.02 | | 2.04 | | 8547 | | 48.1 | | 2.05 | |
| **Dry season** (Total rainfall = 366 mm for Amazon Dataset) | | | | | | | | | | | | |
| 0.5-2 | 73 | | 1.19 | | 1.06 | | 4453 | | 24.9 | | 0.08 | |
| 2-4 | 33 | | 2.79 | | 1.32 | | 4694 | | 30.9 | | 0.15 | |
| 4-6 | 24 | | 4.76 | | 1.29 | | 7417 | | 33.1 | | 0.27 | |
| 6-10 | 31 | | 7.72 | | 1.53 | | 5045 | | 37.7 | | 0.38 | |
| 10-20 | 34 | | 14.92 | | 1.89 | | 4151 | | 42.5 | | 0.65 | |
| 20-40 | 30 | | 28.60 | | 2.13 | | 4275 | | 46.4 | | 1.16 | |
| 40-60 | 14 | | 49.35 | | 2.24 | | 5349 | | 49.3 | | 1.93 | |





**Table 2.** Radar rainfall and self-consistency relations for GoAmazon2014/5 dataset, for the cumulative dataset All, Wet, and Dry seasons, as well as convective and stratiform precipitation as based on RWP classifications. Coefficients estimated at S, C, and X-band radar wavelengths.

| Wavelength | | R(Z) (T=20°C) | R($K_{DP}$) (T=20°C) | R(A) (T=20°C) | R(A) (T=10°C) |
|---|---|---|---|---|---|
| S band | All | $Z = 343.9R^{1.4}$ | $R = 54.2K_{DP}^{0.8}$ | $R = 2904.2A^{1.0}$ | $R = 2227.6A^{1.0}$ |
| | Wet season | $Z = 329.5R^{1.4}$ | $R = 55.2K_{DP}^{0.8}$ | $R = 2949.6A^{1.0}$ | $R = 2265.1A^{1.0}$ |
| | Dry season | $Z = 388.3R^{1.4}$ | $R = 51.5K_{DP}^{0.8}$ | $R = 2732.3A^{1.0}$ | $R = 2090.5A^{1.0}$ |
| | Convective | $Z = 339.9R^{1.4}$ | $R = 54.6K_{DP}^{0.8}$ | $R = 2895.0A^{1.0}$ | $R = 2219.6A^{1.0}$ |
| | Stratiform | $Z = 385.8R^{1.4}$ | $R = 51.1K_{DP}^{0.8}$ | $R = 2867.1A^{1.0}$ | $R = 2202.0A^{1.0}$ |
| C band | All | $Z = 289.0R^{1.4}$ | $R = 30.6K_{DP}^{0.8}$ | $R = 287.8A^{0.9}$ | $R = 239.4A^{0.9}$ |
| | Wet season | $Z = 280.6R^{1.4}$ | $R = 31.3K_{DP}^{0.8}$ | $R = 314.4A^{0.9}$ | $R = 258.3A^{0.9}$ |
| | Dry season | $Z = 314.8R^{1.4}$ | $R = 28.5K_{DP}^{0.8}$ | $R = 242.1A^{0.9}$ | $R = 203.4A^{0.9}$ |
| | Convective | $Z = 281.6R^{1.4}$ | $R = 30.7K_{DP}^{0.8}$ | $R = 278.4A^{0.9}$ | $R = 232.3A^{0.9}$ |
| | Stratiform | $Z = 339.8R^{1.4}$ | $R = 29.5K_{DP}^{0.8}$ | $R = 290.6A^{0.9}$ | $R = 239.7A^{0.9}$ |
| X band | All | $Z = 261.4R^{1.6}$ | $R = 21.5K_{DP}^{0.8}$ | $R = 41.4A^{0.8}$ | $R = 43.0A^{0.8}$ |
| | Wet season | $Z = 239.1R^{1.6}$ | $R = 21.6K_{DP}^{0.8}$ | $R = 42.7A^{0.8}$ | $R = 43.9A^{0.8}$ |
| | Dry season | $Z = 303.2R^{1.6}$ | $R = 21.1K_{DP}^{0.8}$ | $R = 38.2A^{0.8}$ | $R = 40.0A^{0.8}$ |
| | Convective | $Z = 250.2R^{1.6}$ | $R = 21.6K_{DP}^{0.8}$ | $R = 41.0A^{0.8}$ | $R = 43.0A^{0.8}$ |
| | Stratiform | $Z = 318.5R^{1.6}$ | $R = 19.6K_{DP}^{0.8}$ | $R = 40.8A^{0.8}$ | $R = 41.3A^{0.8}$ |

| Self-consistency (T=20°C) | | |
|---|---|---|
| S band | All | $Z = 45.6 + 10.04\log(K_{DP}) + 3.20Z_{DR}$ |
| | Wet season | $Z = 45.7 + 10.10\log(K_{DP}) + 3.17Z_{DR}$ |
| | Dry season | $Z = 45.6 + 10.05\log(K_{DP}) + 3.16Z_{DR}$ |
| C band | All | $Z = 43.3 + 10.12\log(K_{DP}) + 1.96Z_{DR}$ |
| | Wet season | $Z = 43.3 + 10.18\log(K_{DP}) + 1.94Z_{DR}$ |
| | Dry season | $Z = 43.4 + 10.12\log(K_{DP}) + 1.82Z_{DR}$ |
| X band | All | $Z = 38.6 + 9.54\log(K_{DP}) + 4.62Z_{DR}$ |
| | Wet season | $Z = 38.7 + 9.54\log(K_{DP}) + 4.52Z_{DR}$ |
| | Dry season | $Z = 38.7 + 9.80\log(K_{DP}) + 4.89Z_{DR}$ |





**Table 3.** Total convective precipitation and convective fraction for the GoAmazon2014/5 campaign for the cumulative dataset All, Wet, Dry and Transitional seasons. Additional breakdowns for proxy deep convection (ETH > 9 km) and congestus (ETH < 9 km) clouds. Values are estimated according to segregations following BR methods, a hybrid BR/TM combination, the RWP classification, and a simple rainfall rate $R > 10 \ \mathrm{mmhr}^{-1}$ threshold.

| | | BR | BR/TM | RWP | R>10 mmhr$^{-1}$ |
|---|---|---|---|---|---|
| | All | 1378.9 (73.3) | 1584.0 (84.2) | 1565.7 (83.2) | 1480.7 (78.7) |
| Total convective | > 9 km | 765.3 (75.6) | 838.3 (82.8) | 820.7 (81.0) | 801.2 (79.1) |
| precipitation [mm] | < 9 km | 613.5 (70.7) | 745.7 (85.9) | 745.1 (85.8) | 679.5 (78.3) |
| (Convective | Wet season | 664.1 (72.3) | 784.5 (85.4) | 759.2 (82.7) | 709.5 (77.3) |
| fraction [%]) | Dry season | 232.7 (85.7) | 245.2 (90.3) | 255.5 (94.1) | 240.6 (88.6) |
| | Transitional season | 482.1 (69.8) | 554.2 (80.2) | 551.0 (79.7) | 536.2 (77.6) |