# Peer review of "The Green Ocean: Precipitation Insights from the GoAmazon2014/5 Experiment"

_Atmospheric Chemistry and Physics, 2018_

## Referee Comment (RC1) · Anonymous Referee #1 · 16 Apr 2018

Overall: This paper looks at precipitation characteristics during the GoAmazon2014/5 campaign using coupled RWP and disdrometer measurements. The study has the aims of improving our 1) interpretation of radar rainfall relationships, 2) understanding of DSD differences between wet/dry seasons and convective/stratiform regimes, 3) understanding of the possible role of polluted/clean regimes in invigorating convection, and 4) concept of the 'Green Ocean' with the previous points taken as context. There are many angles to this study, and overall it does a good job of addressing the above points. The language and flow of the paper, along with how 'in-depth' the discussion was, greatly improved as the paper progressed; I think it would be beneficial to flesh out some of the discussion in sections 3-4, improve the structure/flow of some of the paragraphs, and clean up some of the figures.

[Figure]

Recommendation: Accept with minor revisions/considerations.

Introduction

This section successfully highlights the motivations and benefits of studying precipitation properties using data from the GoAmazon2014/5 campaign; however, certain sentences/paragraphs seem to be more elaborate repetitions of previous ones (see the following in-depth comments for examples). Concisely rephrasing some parts may make this section less wordy and easier to parse through. Overall, it does do a good job of summarizing the many different angles/benefits this study will have.

P2 L2-6 "diverse forcing conditions" is vague; sentences could be restructured, as it seems a bit circular/hard to follow

P2 L9-11 This is a bit repetitive of the previous paragraphs; could integrate together somehow.

P2 L11 "Low-level barrier" is a strange phrase to use?

P2 L13-15 Needs restructuring/rewording, seems out of place as it is now

P2 L18 Mention that this is what was used in GoAmazon?

P2 L23-25 The "Green Ocean" terminology needs to be better defined

P2 L22-32 These set of sentences need to be restructured or rephrased, especially towards the latter lines

P2 L34-35 I am not sure what is being said in this sentence and would suggest rewording

P3 L4-5 This study seems pretty focused on just the GoAmazon dataset; is it accurate to say that you are looking at this in the context of global variability and shifts in this?

P3 L5 What is the benefit/meaning behind "(or, practical)"?

P3 L10 Unnecessary "a" in ". . .capture a continuous convective cloud. . ."

Dataset and Methodology

I liked this section, 2.1 in particular gives a great overview of the total precipitation statistics embedded in a realistic idea of the associated uncertainties. The paper covers a lot, and the use of specific subsections made sure the reader was orientated properly. The main confusion for me is within the discussion for Figure 1, which could possibly use some clarification (see in-depth comments).

2.1 The ARM T3 Precipitation Dataset and Processing

Subtitle (and subsequent subtitles) Unnecessary "the"; this subsection also concerns the RWP, so not just the precipitation dataset?

P3 L29 Reference/reason for R>0.5 mmhr^(-1) and total drops > 100?

P4 L2 Missing "and" before listing the mass-weighted mean diameter

P4 L9 Comma missing after (degkm-1)

P4 L10 Why 20 degreesC?

P4 L18 Is this from your RWP analysis? Or should there be a reference here?

P4 L20-22 This seems like an incomplete sentence as you state "we first…" and do not follow it with anything else

2.2 The Arm T3 Aerosol Observations and Aerosol Regime Classification

P4 L30 "…aerosol classification is that each…"

P5 L4 The phase "and associated classifications" seems unnecessary

P5 L5 "…potentially a factor of 3 or more…"

P5 L10-12 Are these from Thalman et al. (2017) again?

2.3 PARSIVEL Sampling and Rainfall Relationship Interpretation

P5 L21-22 Uncertainty in what? Radar quantities? Is there a way to provide a numerical value for the "variability established by previous studies" here?

P5 L22 Unnecessary "However,". Does the anticipated estimated uncertainty mentioned previously not include instrument offsets? These two sentences don't flow very well and could possibly do with some rearranging

P5 L27-30 It may be beneficial to add a reference as to why 'b' is fixed and the variability of 'a' is tested P5 L34 (Figure 1) For my clarification, Figure 1 is for the S-band wavelength, yet the discussion of the figure seems to include that of shorter wavelengths in comparison? Am I misinterpreting something here?

Summary Precipitation Results and Interpretation for Retrieval Methods

I enjoyed the discussion on the wet/dry seasonal differences. I feel like this is one of the most interesting parts of this section, but that some explanations and discussion of observations could be expanded upon.

P6 L9-11 Have you considered/is it possible to compare or mention possible datasets from a climate/region more comparable to the Amazon? SGP will obviously (and as acknowledged by yourselves) have differences to the Amazon even though it is continental, so do any datasets (even short term) exist for an even more relevant point of comparison? I do like how they both have similar processing, so I understand why this dataset is used and do not think it is unnecessary, but extra comparisons may be interesting

Table 1 The AM and SGP abbreviations could be defined in the caption for completeness

P6 L7-16 I understand that you have addressed the reasons why there are differences elsewhere, but a sentence on the fact that a tropical vs midlatitude location would account for these differences may be worthwhile here. A lot of what is said here is expanded on in later sections – is this paragraph completely necessary?

Table 2 Is this better suited in supplemental material?

P6 L21-22 SGP is a continental reference, but I think more emphasis should be on the comparison between latitudes instead

Figure 2 Is there any way to show wavelength differences on the same/additional figure? Several parts of the discussion reference wavelength differences and it may be nice to see them summarized in a figure

P6 L30-31 Would be nice to expand on this "significant change" a little here – what/why etc.

Convective/Stratiform Regimes for Rainfall Relationships and DSD Properties

P7 L25 Have the authors considered restructuring to explain the convective/stratiform classification before analyzing the results of doing such here? I cannot find where the RWP classification is actually discussed. A few sentences of what the classification would be beneficial to the reader

Figure 3 This figure could be cleaned up. Consider moving the color bar for the top/bottom two plots outside of the main displays so it is less cluttered and can be seen more clearly.

Figure 4 The white color assigned to the low probabilities is not easy to see when there is a white background. Furthermore, the color bar for these probabilities seems odd – is there any other way to present the probability legend (same problem with subsequent figures)? To make the captions clearer, I would address the shading and the dots separately in terms of what they represent

Amazon Precipitation Properties: Cumulative Dataset Characteristics

The differences between wet and dry seasons could be highlighted more in this section since it is an important aspect of the study. The findings on precipitation originating from congestus/shallower cloud systems is nontrivial, and the importance of this in

context of the introduction could be emphasized more. I think explaining the RWP classification before now will help in latter parts of this section

Figure 5 I like this figure, but the legend/labels in subplot c need to be fixed as the overlapping text is confusing/messy

P8 L22 The fact that cloud top heights could extend 2 km above the heights shown seems a large difference. Can some comment be made about whether this is significant? Does this have any implications for the study or conclusions?

P8 L22-24 What is the reasoning behind relating the winds during the wet/dry season to the modest values of Z in these sentences? Not a lot is said about subplot 5c and the need for highlighting the wind directions in the different seasons

P9 L3 What is meant by "magnitude exceeding threshold as on Figure 6"?

4.1 Disdrometer Convective-Stratiform Segregation: Alignment with RWP Signatures

P9 L29-30 I agree that the use of BR would minimize the contributions of shallower organized convection – but I am unsure of the applications of this given that a large portion of deeper organized convection also falls to the left of the BR line

Figure 8 Should re-label the dashed and solid lines (BR/TM)

P10 L3 I am not sure what the phrase "orients this dataset in BR/TM formulation spaces to the left of the BR line" means. Also, more explanation of how this figure was produced may be necessary.

P10 L5-6 Looking at Figures 8a/c, it isn't obvious that contributions to the histogram are oceanic. I.e. DSDs are not significantly above the dashed TM line. The argument looks more solid for those DSDs identified as stratiform by the RWP

4.2 Cumulative Precipitation Properties According to Cloud Regime and Season

P10 L11 Again, need some discussion previous to this to back up why this is assumed

to be a reasonably proxy

P10 L14-15 "Deeper cumulus clouds are associated with additional maritime continental DSD properties as similar to Darwin studies" – I look at Figure 9a, and I am not sure I see where this comes from. It is not obvious to me that a significant portion is to the right of the BR line, at least not relative to Figure 9b.

4.3 Stratiform Precipitation Properties Associated with Amazon Convective Events

P11 L2 An expansion on the discussion on Table 3 should either be given, or perhaps move this table to supplemental material. Not much is said about it here and it gets overlooked

Amazon Precipitation Properties: The Green Ocean Characteristics

Figure 13 Nothing is written in the text about the transitional period.

P14 L8 "...role of aerosols in the following invigoration arguments"

---

## Referee Comment (RC2) · Anonymous Referee #2 · 25 Apr 2018

**The Green Ocean: Precipitation Insights from the GoAmazon2014/5 Experiment**

**Summary**

The authors explore some of the precipitation events during the wet and dry season of the GoAmazon experiment, and try to decouple the aerosol/thermodynamic characteristics of the precipitation in each regime, and whether or not and the extent to which the precipitation exhibits an oceanic "flavor," as the Amazon basin is often dubbed the "Green Ocean."

The debate about aerosol vs. thermodynamic effects, especially with regards to the intensity of convection and the DSDs of convective regimes continues to rage, and the authors rightly note that the two effects are tightly coupled and it is difficult to disentangle the effects from each other.

Still, this is thorough investigation and a nice summary of the precipitation tendencies of the convection over the Amazon, and provides insight into the meteorological and instrumental goings-on of the GoAmazon campaign. I have one minor point to pick about the CAPE/CIN statistics presented. I also add a thorough list of line-by-line comments but they are largely about writing and style. Overall, very minor revisions.

**Decision:** Accept, with minor revisions.

**Comments**

**Major Comments:**

With particular regard to the statistical analysis in lines 17-23 on page 13, as the distributions of CAPE and CIN are not normal distributions, I feel like a lot of detail is lost in presenting the mean and standard deviations alone. Especially if I am to accept the comparisons made throughout this paragraph, when the differences in MUCAPE (for example) for the different subsets are only a few hundred J/kg/K apart, but the standard deviations are over a thousand, or for the MUCIN, where one standard deviation away from the mean is of the opposite sign, I find those comparisons to be shaky. Could perhaps you show us histograms of the distributions themselves? Or instead present quartiles, or 10%, 90% quantiles? I feel like those would be more representative of these populations.

The figures in general are beautifully done. I think Figures 10, 13, and 15 might be easier to read if presented in one or two vertical columns, and the plot space of each panel increased. Similarly, it would be nice if the plot space of Figure 4b/c could be increased, as currently the detail of the blue contours is completely lost.

Throughout the text, there is a propensity to attribute verbs to the figure titles, as in "Figure 2 plots. . ." or "Figure 2 overlays. . . " While I take no issue with the instances in which you say "indicates," I reacted very strongly to the other instances where you are attributing action words to the figures themselves. I found it to be very distracting, and I suggest revising this throughout the text.

**Minor/Line-by-Line Comments:**

**1.16** "to better inform" is a split infinitive, and while manuals of style say that it has become perfectly acceptable to split an infinitive when justifiable, there are several of them throughout the text. Perhaps diversifying the verbiage a little throughout the manuscript would help with the readability. This is a pedantic point, and you can keep the phrasing as-is, at your discretion.

**2.4** Suggest " . . . *improving model capabilities* introduces *new challenges . . .*"

**2.7** do "observations" and "modeling" need to be capitalized?

**2.7** "were motivated"

**2.10** "*future improvement* of *GCMs*"

**2.11** "low-level barrier" when talking about circulation and dynamics is making me think of dynamic phenomena and not observation obstacles - can you rephrase?

**2.22** "are" does not agree with the subject "perspective" - this entire clause could use some rearranging for clarity.

**2.24** Suggest " . . . *regional characteristics* of convection over the Amazon *that spans oceanic . . .*"

**2.26** Suggest " . . . *but* may *also experience a range of* thermodynamic *and aerosol . . .*"

**2.29** I believe the sentence should read "*the prevalence . . .* is *underappreciated . . .*" but also, the second clause of this sentence is confusing, suggest rephrasing.

**2.34** Suggest "in order to identify" instead of "towards identifying"

**2.35** Suggest "trends" instead of "adheres"

**2.35** Suggest adding more text to "*oceanic, maritime to continental characteristics.*" Maybe something like " . . . precipitation sampled in the Amazon basin trends more towards ocean, maritime characteristics, and when it trends towards possible continental characteristics." This might sound redundant but I had a hard time understanding where the logic of this sentence was going.

**3.9** Suggest *"... and* the *possible effects of the Manaus, Brazil pollution plume."*

**3.10** Suggest removing "a" from "a continuous convective cloud ... "

**3.13** Suggest removing " ... *useful for future hydrological applications"* I understand you are putting in the motivation for that section, but to say it here disrupts the flow of the sentence.

**3.23** Suggest "a *period*"

**4.27** Suggest "the *number concentration*". Also the phrase "*particles condensation nuclei*" is confusing.

**4.29** suggest "other *supplemental materials"*

**5.3** suggest "values of"

**5.5** suggest " ... *potentially* a 3 or more factor of difference ..."

**5.30** onwards - suggest potentially writing coefficients as italicized letters, either by themselves, or with -coefficient, but the quote marks are awkward.

**5.33** sensitivity to what?

**6.4** " ... *basic interpretations* of *the significant changes ..."*

**6.11** "similarly"

**6.14** why the hyphen in "higher-relative"

**6.18** "*Figure 2 plots ..."* this is the first instance of the Figure-verb issue I discussed above. I won't point out each of them, but this is an example of what I struggled with. I suggest instead something like "In Figure 2, we plot summary dataset scatterplots overlaid with dual-polarization relationship fits."

**7.16** Suggest " ... *convective environments* that favor *enhanced evaporation, cooling and subsidence,* which are *less capable* of sustaining ... "

**7.28** why is "For" capitalized in "For example"?

**8.5** what is less influenced? What is the object of this clause?

**8.10** I'd suggest just saying "higher extreme parameter spaces" instead of saying "select" and then saying what it is anyhow.

**8.30** suggest removing comma after "loosely"

**9.10** suggest "The results in Figure 6 . . ."

**9.25** Suggest " . . . *included modest convective diversity*, including *congestus clouds*, and clouds with *maritime, continental, and deeper convective properties (those supporting additional graupel growth)."*

**9.27** suggest " . . . *conditions than* what is *observed* over the *Amazon."*

**9.31** suggest " . . . *limitations for* imposing *BR concepts* when *characterizing . . ."*

**9.33** can you put "*herein TM*" within the parentheses?

**10.4** suggest "either" instead of "belonging to"

**10.14** suggest " . . . *as* is *similar to . . ."*

**10.18** " . . . *having corresponding stratiform DSDs (or, the absence thereof) . . ."* seems contradictory. If having a corresponding stratiform DSD is indicative of TM oceanic characteristics, why then would also its absence?

**10.22** you say "adjacent" and then describe "transitional" - are they the same? and if so, can you pick one?

**11.4** "a *bright band signature*"

**11.18** suggest removing "*are those that*"

**12.2** Suggest excising the sentence "*A more practical . . . GoAmazon2014/5."* it's awkward here.

**12.14** the letters in "convective available potential energy" don't need to be capitalized.

**12.16** suggest removing "that" from "*studies that indicate*"

**13.1** suggest ". . . *drops,* and toward *parameter spaces . . ."*

**13.3** "*though* they *do suppor*t"

**13.4** perhaps change title to "Role of Pollution **in** Oceanic Signatures" ?

**13.12** suggest "The *rightmost panels* of *Figure 15* show *a composite mean . . ."*

**13.16** remove *", respectively*"

**13.17-23** Here is the paragraph in which I struggled with the presentation of representative statistics.

**14.5** "*One explanation . . .* is that *more prominent . . . contributions* are *acting within these convective columns . . .*"

**14.16** " *. . . suggest* that *cleaner aerosol conditions* are *associated . . .*"

**14.31** suggest " *. . . wind directions*, and therefore *should not be as influenced . . .* "

**14.33** suggest *"(e.g., local sources)"*

**15.19** " *. . . explanations for* why *these outliers cluster . . ."*

**15.29** suggest " *. . .initiation* and *subsequent precipitation . . ."*

**15.33** change "were" to "was"

**15.35** suggest "tended to be associated"

**16.11** suggest removing "properties"

**16.16-onward** - use caution with throwing the word "storm" around. You haven't defined what a storm is, and I'm guessing you don't mean a thunderstorm. You don't use this word until this last section. I'd suggest saying "event" from here onward, just to be safe.

**16.22** "*. . . continental behaviors* as seen in *previous studies . . ."*

**16.26** swap "Consulting" with "Considering"

**16.28** remove "radiosonde" - the balloons aren't doing the forcing (although that would be some wild micro-scale meteorology!)

**16.29** suggest "*character* of *the congestus"*

**16.30** suggest "*segregating* by *wind direction"*

**17.3** "a *topic of future consideration"*

---

## Author Comment (AC1) · 15 Jun 2018

**Response to Anonymous Referees,**

**The Green Ocean: Precipitation Insights from the GoAmazon2014/5 Experiment**

**Die Wang et al.**

The authors would like to thank all reviewers for their helpful comments and suggestions. We have responded to all reviewers in a single document. As a brief summary, the revisions to the manuscript include the following highlights:

- We have modified several of the previous images (fonts, lines, sizing, labels, etc.)
- The manuscript has incorporated several changes in response to reviewer comments.
- Additional supplemental materials have been provided, including six figures, and used to relocate previous Table 3.

The individual reviewer comments and responses are included in the following document (author comments in black, reviewer comments in grey and italics).

**Response to Anonymous Referee #1**

*Overall: This paper looks at precipitation characteristics during the GoAmazon2014/5 campaign using coupled RWP and disdrometer measurements. The study has the aims of improving our 1) interpretation of radar rainfall relationships, 2) understanding of DSD differences between wet/dry seasons and convective/stratiform regimes, 3) understanding of the possible role of polluted/clean regimes in invigorating convection, and 4) concept of the 'Green Ocean' with the previous points taken as context. There are many angles to this study, and overall it does a good job of addressing the above points. The language and flow of the paper, along with how 'in-depth' the discussion was, greatly improved as the paper progressed; I think it would be beneficial to flesh out some of the discussion in sections 3-4, improve the structure/flow of some of the paragraphs, and clean up some of the figures.*

**We thank the reviewer for their kind words. We hope our revisions are sufficient to address many concerns of this reviewer.**

*Introduction*
*This section successfully highlights the motivations and benefits of studying precipitation properties using data from the GoAmazon2014/5 campaign; however, certain sentences/paragraphs seem to be more elaborate repetitions of previous ones (see the following in-depth comments for examples). Concisely rephrasing some parts may make this section less wordy and easier to parse through. Overall, it does do a good job of summarizing the many different angles/benefits this study will have.*

*P2 L2-6 "diverse forcing conditions" is vague; sentences could be restructured, as it seems a bit circular/hard to follow*

**Agree with the reviewer. Replacing the line, "Thus, improving precipitation measurements (i.e., those that better reflect the natural variability of clouds under diverse forcing conditions) has traditionally supported improved convective treatments."**

**With,**

**"Thus, a traditional observational approach in support of convective modeling has been to document global precipitation variability and improve basic rainfall retrievals."**

*P2 L9-11 This is a bit repetitive of the previous paragraphs; could integrate together somehow.*

**Agree. Can drop the line (and references), "The inability of GCMs to adequately represent cumulus clouds and precipitation over the Amazon highlights one example of the observational needs for future improvement to GCM cloud parameterizations and the larger-scale circulation connections therein (e.g., Richter and Xie, 2008; Nobre et al., 2009; Yin et al., 2013)"**

*P2 L11 "Low-level barrier" is a strange phrase to use?*

**Agree. Replaced line with, "One source of uncertainty when developing useful precipitation retrievals for model development is the shortage of long-term surface gauge and disdrometer observations within tropical regions."**

*P2 L13-15 Needs restructuring/rewording, seems out of place as it is now*

**Agree. Replaced, "Geophysical retrievals of interest for precipitation studies include radar-based rainfall estimation, but even basic radar preprocessing improvements for dual-polarization quantities can be critical for future studies utilizing forward radar model comparisons."**

**With,**

**"Although radar rainfall estimation and its uncertainty for tropical applications is of primary interest, basic radar preprocessing, calibration and dual-polarization radar data quality is also improved with extended surface precipitation records in diverse environments."**

*P2 L18 Mention that this is what was used in GoAmazon?*

**Agree. Changed the line to, "Establishing boundaries for tropical precipitation expectations and radar data quality concepts (e.g., Scarchilli et al., 1996, self-consistency methods) provides an immediate benefit when interpreting remote radar deployment datasets including those from the Atmospheric Radiation Measurement (Ackerman and Stokes, 2003, ARM) Mobile Facility (Miller et al., 2016, AMF) during GoAmazon2014/5."**

*P2 L23-25 The "Green Ocean" terminology needs to be better defined*
*P2 L22-32 These set of sentences need to be restructured or rephrased, especially towards the latter lines*

**Revised the paragraph as, "Although improving hydrological retrievals is of a practical significance, an interesting outcome from previous Amazon studies is the labelling of the Amazon as the 'Green Ocean'. This Green Ocean terminology is rooted in studies such as Roberts et al. (2001) wherein low cloud condensation nuclei (CCN) concentrations and high CCN to condensation nuclei (CN) ratios over the Amazon resembled marine environments, as distinct from previous continental expectations. However, this terminology is often extended to include the unique regional characteristics observed from Amazon convection that spans oceanic to continental cloud extremes in key attributes such as updraft intensities and propensity for electrification. Specific to convection, Amazon clouds may initiate under these clean (or lower) CCN conditions, over a pristine forest, but also experience a range of thermodynamical and aerosol forcing influences that promote changes in cloud properties including electrification, cloud droplet size distribution and precipitation changes, or enhanced updraft intensity (e.g., Williams et al., 2002; Cecchini et al., 2016; Giangrande et al., 2016b, 2017). As described by Williams et al. (2002), the prevalence of maritime convective cloud regimes over a large continent are possibly still underappreciated in the convective cloud spectrum and its intensity, especially given the propensity to identify deeper convection over the Amazon having electrification arguing continental convective characteristics."**

*P2 L34-35 I am not sure what is being said in this sentence and would suggest rewording*

**Reworded to, "One motivation for this study is to identify conditions under which precipitation sampled in the Amazon basin adheres more towards oceanic, maritime and continental characteristics (e.g., Tokay and Short, 1996)."**

*P3 L4-5 This study seems pretty focused on just the GoAmazon dataset; is it accurate to say that you are looking at this in the context of global variability and shifts in this?*

*P3 L5 What is the benefit/meaning behind "(or, practical)"?*

**Here, reworded the line, "In addition to better addressing Amazon precipitation in the context of global variability, it is useful to assess whether traditional (or, practical) radar remote-sensing (including dual-polarization) quantities are sensitive to these shifts. "**

**To,**

**"While it is important to view these Amazon datasets and cloud or larger-scale regime shifts in the context of global disdrometer observations (e.g., Dolan et al. 2018), it is also useful to determine whether common remote-sensing platforms (e.g., dual-polarization quantities as from X-band to S-band radars) are sensitive to these differences."**

*P3 L10 Unnecessary "a" in "…capture a continuous convective cloud…"*

**Fixed, thanks.**

*Dataset and Methodology*

*I liked this section, 2.1 in particular gives a great overview of the total precipitation statistics embedded in a realistic idea of the associated uncertainties. The paper covers a lot, and the use of specific subsections made sure the reader was orientated properly. The main confusion for me is within the discussion for Figure 1, which could possibly use some clarification (see in-depth comments).*

*2.1 The ARM T3 Precipitation Dataset and Processing*
*Subtitle (and subsequent subtitles) Unnecessary "the"; this subsection also concerns the RWP, so not just the precipitation dataset?*

**Revised to 'ARM T3 Precipitation and Radar Wind Profiler Dataset and Processing'**

*P3 L29 Reference/reason for R>0.5 mmhr-1 and total drops > 100?*

**The drop count and rainfall rate thresholds help prevent small sample sizes from skewing DSD estimates (Smith et al. 1993; Smith 2016; Dolan et al. 2018). These thresholds are typically similar across most of the disdrometer literature (usually following Tokay recommendations, etc.), but could vary if the disdrometer dataset is processed for other time resolutions or for different locations sampling different types of precipitation. These values may also be different if the disdrometer is assumed of higher quality, e.g., 2D Video Disdrometers may allow for lower rainfall rates and/or drop count threshold owing to improved small drop size sampling. For example, in Park et al. (2017), this study processed the 1-min DSDs by removing data if the total number of drops was < 30 and R < 0.01 mmhr$^{-1}$. In regimes with high concentrations of smaller drops (e.g., orographic settings, drizzle), one may potentially implement an even higher drop count threshold.**

**Smith, P. L., Z. Liu, and J. Joss, 1993: A study of sampling-variability effects in raindrop size observations. J. Appl. Meteor., 32, 1259–1269**

**Smith, P. L., 2016: Sampling issues in estimating radar variables from disdrometer data. J. Atmos. Oceanic Technol., 33, 2305–2313**

Dolan, B., B. Fuchs, S.A. Rutledge, E.A. Barnes, and E.J. Thompson, 2018: Primary Modes of Global Drop Size Distributions. J. Atmos. Sci., 75, 1453–1476

Park, S., H. Kim, Y. Ham, and S. Jung, 2017: Comparative Evaluation of the OTT PARSIVEL2 Using a Collocated Two-Dimensional Video Disdrometer. J. Atmos. Oceanic Technol., 34, 2059–2082

*P4 L2 Missing "and" before listing the mass-weighted mean diameter*

**Fixed, thanks.**

*P4 L9 Comma missing after (degkm$^{-1}$)*

**Fixed, thanks.**

*P4 L10 Why 20 degrees C?*

**We adopted 20C as it reflects a common assumption from the literature (to assist in relative alignment, comparisons). This assumption does not impact the estimates of Z or $K_{DP}$ as significantly as it would $A_h$ and relative relations therein. For $A_h$, we provide an additional reference for 10C, as is often reported for multiple temperatures in the literature, e.g., Ryzhkov et al. (2014), Diederich et al. (2015). If the reviewer (or readers of this response) is interested in additional temperature variability, those calculations are not difficult can be performed on request. The eventual ARM disdrometer Value Added Product (VAPs) releases from these efforts will also include a wider range of calculations than what is presented in this manuscript.**

Diederich, M., A. Ryzhkov, C. Simmer, P. Zhang, and S. Trömel, 2015: Use of Specific Attenuation for Rainfall Measurement at X-Band Radar Wavelengths. Part I: Radar Calibration and Partial Beam Blockage Estimation. J. Hydrometeor., 16, 487–502.

Ryzhkov, A., M. Diederich, P. Zhang, and C. Simmer, 2014: Potential Utilization of Specific Attenuation for Rainfall Estimation, Mitigation of Partial Beam Blockage, and Radar Networking. J. Atmos. Oceanic Technol., 31, 599–619.

*P4 L18 Is this from your RWP analysis? Or should there be a reference here?*

**Yes, this statement comes from our RWP analysis (see the figure below). This figure shows the mean Vertical Velocity profiles (in color) as a function of 10 dBZ Echo-Top Height (ETH). This plot is only showing convective locations as observed during GoAmazon2014/5. The ETH (x-axis) and actual height above the RWP (y-axis) line up reasonably, in that the mean convective cloud vertical velocity approaches 0 m/s at this relative altitude (falls along a 1-to-1 line to ~ 15 km).**

[Figure]

*P4 L20-22 This seems like an incomplete sentence as you state "we first…" and do not follow it with anything else*

**Thanks. We have modified the sentence, "For echo classifications, we first identify higher confidence convective and stratiform regions on the basis of column Z signatures and/or so-called radar 'bright band' (melting-level) designations for longer wavelengths (e.g., Fabry and Zawadzki, 1995; Geerts and Dawei, 2004)."**

**To,**

**"For echo classifications, we identify convective and stratiform regions on the basis of column Z signatures, velocity properties, and/or so-called radar 'bright band' (melting-level) designations for longer wavelengths (e.g., Fabry and Zawadzki, 1995; Williams et al. 1995; Geerts and Dawei, 2004)."**

*P4 L30 "…aerosol classification is that each…"*

**Fixed, thanks.**

*P5 L4 The phase "and associated classifications" seems unnecessary*

**OK. We removed it.**

*P5 L5 "…potentially a factor of 3 or more…"*

**Agree. Thanks.**

*P5 L10-12 Are these from Thalman et al. (2017) again?*

**Yes.**

*P5 L21-22 Uncertainty in what? Radar quantities? Is there a way to provide a numerical value for the "variability established by previous studies" here?*

*P5 L22 Unnecessary "However,". Does the anticipated estimated uncertainty mentioned previously not include instrument offsets? These two sentences don't flow very well and could possibly do with some rearranging*

**Rephrase to: "Given our comparisons between rainfall accumulations with surface gauge measurements under typical storm intensities, as well as previous side-by-side performance testing of other PARSIVEL units, we do not anticipate radar quantity uncertainty falling outside the variability established by previous studies. For example, reasonable instrument offsets for radar quantities such as Z may be on the order of 10-20 % or 1-2 dBZ."**

*P5 L27-30 It may be beneficial to add a reference as to why 'b' is fixed and the variability of 'a' is tested*

**Ok. Have added a reference to:**

**Steiner, M., Smith, J. A. and Uijlenhoet, R. 2004 A microphysical interpretation of the radar reflectivity–rain rate relationship. J. Atmos. Sci., 61, 1114–1131**

**As, "As plotted in Figure 1, we show histograms for a-coefficient values from various single parameter rainfall relationships (radar quantities estimated as in previous sections), assuming a fixed b-coefficient as determined from our complete Amazon dataset for the S-band wavelength. This example highlights the sensitivity in the a-coefficients as estimated from random half-dataset subsets to the complete dataset (vertical black line). Assuming a constant b-coefficient ~1.4 is typically a reasonable assumption to assist in microphysical interpretation from R(Z) relationships for size-controlled conditions (e.g., Steiner et al. 2004)."**

*P5 L34 (Figure 1) For my clarification, Figure 1 is for the S-band wavelength, yet the discussion of the figure seems to include that of shorter wavelengths in comparison? Am I misinterpreting something here?*

**Yes, the discussions here are based on a- and b-coefficient distributions that we calculated for multiple wavelengths. Figure 1 only includes S-band shown as an example. We performed the same analysis for shorter wavelengths as well (see figures below for C and X-band), which are not shown in the manuscript, however we provide them below.**

[Figure]

**Figure S1:** Histograms for a-coefficient values from single parameter rainfall relationships (a) R(Z), (b) R(K_DP), and (c) R(A), calculated using least square method under the assumption of a fixed b-coefficient from random sampling of half of the dataset (5000 times), for the C-band wavelength. The red curves represent the fit Gaussian distribution of a-coefficient. The red vertical lines represent the a-coefficient calculated based on the whole dataset.

[Figure]

**Figure S2: The same as Figure1 but for X-band wavelength.**

We will rephrase the line to,

**"Though not shown in Figure 1, a deterioration in performance at shorter wavelengths is found for R (Z) relationships owing to …. The corresponding plots for C-band and X-band are provided in the supplemental material (Figures S1 and S2)."**

As above, since we are including supplemental material, these images can be included in that format as well.

*P6 L9-11 Have you considered/is it possible to compare or mention possible datasets from a climate/region more comparable to the Amazon? SGP will obviously (and as acknowledged by yourselves) have differences to the Amazon even though it is continental, so do any datasets (even short term) exist for an even more relevant point of comparison? I do like how they both have similar processing, so I understand why this dataset is used and do not think it is unnecessary, but extra comparisons may be interesting*

Having a Darwin component that expanded on Giangrande et al. (2014) ARM JWD insights was an initial plan for this manuscript (e.g., an improved tropical comparison with similar disdrometers as a callback to Tokay and Short, etc.). Unfortunately, at the time of writing, we did not have direct permission to the datasets from Darwin, Australia, e.g., those that have been collected by a higher quality 2DVD unit (approximately 5-year archive as established by Bringi and others). The ARM JWD impact disdrometer archive from Darwin was also a possibility, but that instrument type is usually considered of lower quality for dual-polarization efforts.

We have performed some comparisons with the Darwin datasets and find that the results from Darwin are quite similar to those from the Amazon in terms of applications for the Darwin C-band radars and associated rainfall and processing coefficient needs therein (e.g., deeper convection, monsoonal events). The most pronounced difference is in the lower frequency for these shallower/congestus contributions to the rainfall datasets.

*Table 1 The AM and SGP abbreviations could be defined in the caption for completeness*

OK, rephrased the caption of Table 1: …for All, Wet, and Dry seasons, for the Amazon (MAO) and the Southern Great Plains (SGP) sites". We changed AM to MAO to keep that consistent with the name of the dataset on ARM website.

*P6 L7-16 I understand that you have addressed the reasons why there are differences elsewhere, but a sentence on the fact that a tropical vs midlatitude location would account for these differences may be worthwhile here. A lot of what is said here is expanded on in later sections – is this paragraph completely necessary?*

We prefer to introduce each section with a brief summary, an author preference. We have added at the end of that paragraph, "Discrepancies between SGP and Amazon, as well as Wet/Dry separations, are most pronounced at the higher R consistent with convective cores. This is likely based on the propensity for melting hail in deeper SGP convection and/or larger melting aggregates in stratiform regions trailing convective lines favoring larger drop sizes at the surface."

*Table 2 Is this better suited in supplemental material?*

Possibly – However, we think that it is informative to include Table 2 in the main manuscript. We do agree with this reviewer and the second reviewer that the supplemental material option may be useful for Table 3 and to include other wavelength/important information.

*P6 L21-22 SGP is a continental reference, but I think more emphasis should be on the comparison between latitudes instead*

As to a previous reviewer comment, the 'continental' reference is useful when addressing prior 'Green Ocean' terminology and its applicability therein. We would have preferred to include an additional Darwin reference/contribution if a comparable dataset was available to us (we do not have permission and think this comparison would make a lengthy manuscript longer and detract from some of the other findings). Another option within DOE ARM was the Gan dataset as discussed by Thompson et al. (2015). We felt this dataset was more reflective of an 'oceanic' condition, and best illustrated by referring to the TM classification lines.

*Figure 2 Is there any way to show wavelength differences on the same/additional figure? Several parts of the discussion reference wavelength differences and it may be nice to see them summarized in a figure.*

This is an example where we can include those plots as part of the supplemental materials, e.g., Figures S3 and S4. We will include a line in the text that points to the supplemental materials for these figures.

[Figure]

**Figure S3.** Scatter plots of (a) Z, (b) KDP, and (c) A versus rain rate and overlaid associated relationship fits using least square method for Amazon (MAO, solid lines) and SGP-Oklahoma (SGP, dashed lines) sites, for the C-band wavelength. Density is shown in color.

[Figure]

**Figure S4.** The same as Figure S3, but for X-band wavelength.

*P6 L30-31 Would be nice to expand on this "significant change" a little here – what/why etc.*

**Added a better justification, e.g., larger than that expected from subsampling.**

*P7 L25 Have the authors considered restructuring to explain the convective/stratiform classification before analyzing the results of doing such here? I cannot find where the RWP classification is actually discussed. A few sentences of what the classification would be beneficial to the reader*

**In this case, we introduced the classification in the previous sections, but this may not have been clear. As in the section, the classification follows the previous Giangrande et. al. (2014;2016) RWP offerings, however we have added an extra reference in the above (Section 2.1) to Williams et al. (1995) – this was following a suggestion by C. Williams given the relative similarities of the RWP ideas to previous tropical RWP classification efforts he performed. To note, multiple authors have used very similar classification logic, e.g., identify meteorological from non-meteorological echo, identify convective cores, weaker convection from stratiform using precipitation and velocity signature thresholds, and typically a melting layer / bright band check to identify stronger stratiform signatures. The biggest change in this approach is that it starts by using a fuzzy logic approach as its initial backbone for echo identification – but, we still adopt some sanity checks / thresholds, ground precipitation checks, etc., to ensure reasonable echo identification. These efforts have also been improved, cross-checked with the availability of the collocated W-band (WACR) during GoAmazon2014/5 that is not sensitive to Bragg echo/insects.**

*Figure 3 This figure could be cleaned up. Consider moving the color bar for the top/bottom two plots outside of the main displays so it is less cluttered and can be seen more clearly.*

**Thanks. We have replotted the figure.**

*Figure 4 The white color assigned to the low probabilities is not easy to see when there is a white background. Furthermore, the color bar for these probabilities seems odd – is there any other way to present the probability legend (same problem with subsequent figures)? To make the captions clearer, I would address the shading and the dots separately in terms of what they represent*

**Agree. It should read 'Density' in the place of 'Probability'.**

*The differences between wet and dry seasons could be highlighted more in this section since it is an important aspect of the study. The findings on precipitation originating from congestus/shallower cloud systems is nontrivial, and the importance of this in context of the introduction could be emphasized more. I think explaining the RWP classification before now will help in latter parts of this section.*

**In this instance, we agree. We did not develop the Wet/Dry contrasts with much detail, perhaps since we (as authors) knew the more detailed DSD-driven Wet v Dry discussions would follow. This is also challenge when attempting to pack in the quantity of information while keeping certain concepts (e.g., rainfall relations) separate from more detailed (DSD) discussions. Our first thought is to simply acknowledge this to the reader, e.g., including the line - "Nevertheless, summary rainfall properties skew heavily towards convective designations for all seasons, as reported in Table S1. Seasonal changes will be discussed further in Section 4 in the context of multiparameter DSD breakdowns."**

*Figure 5 I like this figure, but the legend/labels in subplot c need to be fixed as the overlapping text is confusing/messy*

**Agree, we replotted Figure 5.**

*P8 L22 The fact that cloud top heights could extend 2 km above the heights shown seems a large difference. Can some comment be made about whether this is significant? Does this have any implications for the study or conclusions?*

**We include this statement (conservative) because these differences between RWP ETH and true CTH may be important for interpretation by future modeling activities (if those comparisons do not apply an appropriate forward radar simulator that can account for radar sensitivity considerations). In a relative sense, this does not have much of an impact, as our ETH criteria was driven by the naturally-occurring separation/bimodality in the ETH measurements from the RWP that is consistent with expected cloud top modes (e.g., Johnson et al. 1999, etc). However, the authors still wanted to note this potential uncertainty as there is an unknown difference between these ETH and cloud top height (same is true even for cloud radars having much higher sensitivity, on which these statements are rooted – having collocated WACR).**

**Johnson, R.H., T.M. Rickenbach, S.A. Rutledge, P.E. Ciesielski, and W.H. Schubert, 1999: Trimodal Characteristics of Tropical Convection. J. Climate, 12, 2397–2418,**

*P8 L22-24 What is the reasoning behind relating the winds during the wet/dry season to the modest values of Z in these sentences? Not a lot is said about subplot 5c and the need for highlighting the wind directions in the different seasons*

**Agree. Joined a sentence incorrectly. Should be rephrased as,**

**"Sounding-based winds over the T3 site are predominantly easterly (mostly observed during the Dry season) to northeasterly (mostly, Wet season) (Figure 5c). Low-level Z observations (Figure 5d) illustrate that Amazon cumulus are often linked to relatively modest values of Z ~ 35 dBZ."**

*P9 L3 What is meant by "magnitude exceeding threshold as on Figure 6"?*

**e.g., when the Vertical Velocity is greater than certain values (like 1 m/s, 3 m/s, and 5 m/s on the plot). We have removed this as it seems confusing.**

*P9 L29-30 I agree that the use of BR would minimize the contributions of shallower organized convection – but I am unsure of the applications of this given that a large portion of deeper organized convection also falls to the left of the BR line*

**We would agree that the main point is to show that a substantial portion of 'convective' clouds would still have DSDs that fall to the left of that separation line.**

*Figure 8 Should re-label the dashed and solid lines (BR/TM)*

**OK, we have re-labeled BR and TM lines on Figures 8, 9, 10, 11, 13, 15, and 16.**

*P10 L3 I am not sure what the phrase "orients this dataset in BR/TM formulation spaces to the left of the BR line" means. Also, more explanation of how this figure was produced may be necessary.*
*P10 L5-6 Looking at Figures 8a/c, it isn't obvious that contributions to the histogram are oceanic. I.e. DSDs are not significantly above the dashed TM line. The argument looks more solid for those DSDs identified as stratiform by the RWP*

**Agree. This should read,**

**"As plotted in Figure 8, we consider only the DSDs that would fall to the left of the BR separation line (e.g., those that follow a traditional BR stratiform designation). For this figure, the DSDs identified as belonging to convective or stratiform (based on the RWP definitions) are then subset according to the left and right panels, respectively."**

**As for the second comment, one can see from Figure 8a that a relative large portion of RWP-convective DSDs fall into/near Thompson oceanic concepts (smaller drops, higher number concentration). The argument is to highlight that there are Amazon convective and stratiform expectations from DSDs that would otherwise be described as 'stratiform' according to BR. The reviewer is correct in the sense of that we do have several DSDs near/below the Thompson TM line, which may be associated with echoes from transitions from convective to stratiform, etc. The RWP designating those areas as convection may simply indicate that regions aloft exhibited convective signatures or these are transitional regions at the periphery of convective regions. Again, the intent was not that all smaller drop or weaker convective Amazon DSDs would be found above this TM line (that would suggest all Amazon cloud-precipitation as**

oceanic), but that a large % were consistent with oceanic properties as well. We would agree that the two 'stratiform' criteria usually implies DSDs clusters are below the Thompson line.

*P10 L11 Again, need some discussion previous to this to back up why this is assumed to be a reasonably proxy*

OK. Here, this choice is based on the bimodal distribution of ETH from Figure 5 and the associated discussion, fixes therein.

*P10 L14-15 "Deeper cumulus clouds are associated with additional maritime continental DSD properties as similar to Darwin studies" – I look at Figure 9a, and I am not sure I see where this comes from. It is not obvious to me that a significant portion is to the right of the BR line, at least not relative to Figure 9b.*

In these examples, we would agree that there is overlap between the two plots. However, this statement (maritime convective) is not necessarily used to imply that additional convection is to the 'right' of this BR line, but that fewer convective observations are found 'above' the TM 'oceanic' line. However, we do also find that there are larger-drop convective extremes, as well as additional evidence for more mature stratiform DSDs that are favoring aggregates (low $N_w$, higher $D_0$ observations).

*P11 L2 An expansion on the discussion on Table 3 should either be given, or perhaps move this table to supplemental material. Not much is said about it here and it gets overlooked*

As discussed with Table 2, it is possible to move this table (which was provided for statistical completeness) to supplemental materials (e.g., as Table S1). We think this change is reasonable and is more appropriate than moving Table 1.

*Figure 13 Nothing is written in the text about the transitional period.*

We would agree (e.g., Figures 10 and 13) minimal text was provided on the Transitional season (MON), and these images were provided for completeness. The months did tend to reflect transitional behaviors between the Wet and Dry season properties. Interestingly, previous studies by the authors (e.g., Giangrande et al. 2016), as well as storm electrification literature, have suggested that many intense convective cells (as defined by updraft speeds) are sampled during Transitional periods (e.g., October, November). In our examples, the Transitional months do not demonstrate cumulative dataset precipitation behaviors/extremes outside the bounds of the Wet or Dry season properties (which may simply be that outlier examples are also not highlighted well by our cumulative plots).

*P14 L8 "…role of aerosols in the following invigoration arguments"*

Agree. Thanks.

**Response to Anonymous Referee #2**

*The authors explore some of the precipitation events during the wet and dry season of the GoAmazon experiment and try to decouple the aerosol/thermodynamic characteristics of the precipitation in each regime, and whether or not and the extent to which the precipitation exhibits an oceanic "flavor," as the Amazon basin is often dubbed the "Green Ocean." The debate about aerosol vs. thermodynamic effects, especially with regards to the intensity of convection and the DSDs of convective regimes continues to rage, and the authors rightly note that the two effects are tightly coupled and it is difficult to disentangle the effects from each other. Still, this is thorough investigation and a nice summary of the precipitation tendencies of the convection over the Amazon and provides insight into the meteorological and instrumental goings-on of the GoAmazon campaign. I have one minor point to pick about the CAPE/CIN statistics presented. I also add a thorough list of line-by-line comments but they are largely about writing and style. Overall, very minor revisions.*

**We thank this reviewer for the comments and suggestions, and we hope we have improved the revised manuscript in ways that respond to any concerns from this reviewer and the other reviewer comments above.**

*Major Comments:*

*With particular regard to the statistical analysis in lines 17-23 on page 13, as the distributions of CAPE and CIN are not normal distributions, I feel like a lot of detail is lost in presenting the mean and standard deviations alone. Especially if I am to accept the comparisons made throughout this paragraph, when the differences in MUCAPE (for example) for the different subsets are only a few hundred J/kg/K apart, but the standard deviations are over a thousand, or for the MUCIN, where one standard deviation away from the mean is of the opposite sign, I find those comparisons to be shaky. Could perhaps you show us histograms of the distributions themselves? Or instead present quartiles, or 10%, 90% quantiles? I feel like those would be more representative of these populations.*

**This is a valid point, and something the authors have discussed with others who have performed similar aerosol-cloud studies (e.g., Varble 2018). Modest variability in CAPE/CIN can have a significant impact – and that even a few hundred J/kg/K (if that was even a reasonable assumption) is non-trivial. We agree that at a minimum, the authors should provide histograms of these MUCAPE and MUCIN (as for example, in supplemental materials as suggested by reviewer 1). These can be found below and also in Figure S5, with an additional line added to the manuscript to point to this image. We have also added in a set of histograms for the later MUCAPE/MUCIN breakdowns as a function of wind direction in Figure S6 (supplemental, see below image). We think this is also useful to show for the readers/reviewers.**

Varble, A., 2018: Erroneous Attribution of Deep Convective Invigoration to Aerosol Concentration. J. Atmos. Sci., 75, 1351–1368, https://doi.org/10.1175/JAS-D-17-0217.1

[Figure]

**Figure 5S: Histograms of MUCAPE and MUCIN for polluted and clean cases.**

[Figure]

**Figure 6S: Histograms of MUCAPE and MUCIN for different wind directions.**

*The figures in general are beautifully done. I think Figures 10, 13, and 15 might be easier to read if presented in one or two vertical columns, and the plot space of each panel increased. Similarly, it would be nice if the plot space of Figure 4b/c could be increased, as currently the detail of the blue contours is completely lost.*

**Yes, agree. We have modified the plots.**

*Throughout the text, there is a propensity to attribute verbs to the figure titles, as in "Figure 2 plots. . ." or "Figure 2 overlays. . . " While I take no issue with the instances in which you say "indicates," I reacted very strongly to the other instances where you are attributing action words to the figures themselves. I found it to be very distracting, and I suggest revising this throughout the text.*

**Ok, agree. We have revised several 'Figure * plots/presents/overlays/separates' and other instances throughout the text. These can be revised as suggested by the reviewer below.**

*Minor/Line-by-Line Comments:*

*1.16 "to better inform" is a split infinitive, and while manuals of style say that it has become perfectly acceptable to split an infinitive when justifiable, there are several of them throughout the text. Perhaps diversifying the verbiage, a little throughout the manuscript would help with the readability. This is a pedantic point, and you can keep the phrasing as-is, at your discretion.*

Thank you for drawing this to our attention. We note many examples (esp. early on in the manuscript) such as 'to better inform', 'to better constrain', 'to adequately represent', 'To better understand', 'to better addressing', 'to further differentiate', 'to better isolate'. In most cases, we can easily remove the adverb (typically, the word 'better') and our point would be the same.

*2.4 Suggest " . . . improving model capabilities introduces new challenges . . ."*

**Agree. Thanks.**

*2.7 do "observations" and "modeling" need to be capitalized?*

**Here, we think this use "Observations" and "Modeling" is appropriate for ACP as this seems to be the formal campaign name / use by the PI / overview article and other previous literature presents the official campaign naming in the ACP / special edition.**

*2.7 "were motivated"*

**Agree. Thanks.**

*2.10 "future improvement of GCMs"*

**Agree. Thanks.**

*2.11 "low-level barrier" when talking about circulation and dynamics is making me think of dynamic phenomena and not observation obstacles - can you rephrase?*

**Agree. This has been addressed as in following comments to Reviewer 1.**

*2.22 "are" does not agree with the subject "perspective" - this entire clause could use some rearranging for clarity.*
*2.24 Suggest " . . . regional characteristics of convection over the Amazon that spans oceanic . . ." 2.26 Suggest " . . . but may also experience a range of thermodynamic and aerosol . . ."*
*2.29 I believe the sentence should read "the prevalence . . . is underappreciated . . ." but also, the second clause of this sentence is confusing, suggest rephrasing.*

**Agree. Revised this paragraph as in following Reviewer 1 and Reviewer 2 comments.**

*2.34 Suggest "in order to identify" instead of "towards identifying"*

**Accepted. Thanks.**

**As in the Reviewer 1 response, reworded to, "One motivation for this study is to identify conditions under which precipitation sampled in the Amazon basin adheres more towards oceanic, maritime and continental characteristics (e.g., Tokay and Short, 1996)."**

*2.35 Suggest "trends" instead of "adheres"*

**Accepted. Thanks.**

*2.35 Suggest adding more text to "oceanic, maritime to continental characteristics." Maybe something like ". . . precipitation sampled in the Amazon basin trends more towards ocean, maritime characteristics, and when it trends towards possible continental characteristics." This might sound redundant but I had a hard time understanding where the logic of this sentence was going.*

**Accepted. Thanks.**

*3.9 Suggest ". . . and the possible effects of the Manaus, Brazil pollution plume."*

**Accepted. Thanks.**

*3.10 Suggest removing "a" from "a continuous convective cloud . . . "*

**Accepted. Thanks.**

*3.13 Suggest removing ". . . useful for future hydrological applications" I understand you are putting in the motivation for that section, but to say it here disrupts the flow of the sentence.*

**Accepted. Thanks.**

*3.23 Suggest "a period"*

**Accepted. Thanks.**

*4.27 Suggest "the number concentration". Also, the phrase "particles condensation nuclei" is confusing.*

**Rephrased as, "Aerosol regime classifications are based on the combination of condensation nuclei (CN) measurements, measurements for the fraction of particles with diameters less than 70 nm, and carbon monoxide CO and oxides of Nitrogen ($NO_y$) measurements at the T3 location using instrumentation as described in Thalman et al. (2017) and supplemental materials."**

*4.29 suggest "other supplemental materials"*

**Agree. Thanks.**

*5.3 suggest "values of"*

**Agree. Thanks.**

*5.5 suggest ". . . potentially a 3 or more factor of difference . . ."*

**Agree. Thanks.**

*5.30 onwards - suggest potentially writing coefficients as italicized letters, either by themselves, or with -coefficient, but the quote marks are awkward.*

**OK, changed to '*a*-coefficient' *for all the places.***

*5.33 sensitivity to what?*

**Changed to 'sampling uncertainty'**

*6.4 " . . . basic interpretations of the significant changes . . ."*

**Accepted. Thanks.**

*6.11 "similarly"*

**Agree. Thanks.**

*6.14 why the hyphen in "higher-relative"*

**OK. Remove "-"**

*6.18 "Figure 2 plots . . ." this is the first instance of the Figure-verb issue I discussed above. I won't point out each of them, but this is an example of what I struggled with. I suggest instead something like "In Figure 2, we plot summary dataset scatterplots overlaid with dual-polarization relationship fits."*

**Yes, we have revised 'Figure * plot' examples throughout the text.**

*7.16 Suggest " . . . convective environments that favor enhanced evaporation, cooling and subsidence, which are less capable of sustaining . . . "*

**Accepted. Thanks.**

*7.28 why is "For" capitalized in "For example"?*

**OK. Changed that to "for example"**

*8.5 what is less influenced? What is the object of this clause?*

**Agree. Rephrased as, 'This reduced coefficient variability reflects on the closer relationship between A and $K_{DP}$ with rainfall rate, less influenced by the presence/absence of select larger drop sizes.'**

*8.10 I'd suggest just saying "higher extreme parameter spaces" instead of saying "select" and then saying what it is anyhow.*

**Agree. Thanks.**

*8.30 suggest removing comma after "loosely"*

**Accepted. Thanks.**

*9.10 suggest "The results in Figure 6 . . ."*

**Agree. Thanks.**

*9.25 Suggest " . . . included modest convective diversity, including congestus clouds, and clouds with maritime, continental, and deeper convective properties (those supporting additional graupel growth)."*

**Accepted. Thanks.**

*9.27 suggest " . . . conditions than what is observed over the Amazon."*

**Accepted. Thanks.**

*9.31 suggest " . . . limitations for imposing BR concepts when characterizing . . ."*

**Agree. Thanks.**

*9.33 can you put "herein TM" within the parentheses?*

**Done. Thanks.**

*10.4 suggest "either" instead of "belonging to"*

**Accepted. Thanks.**

*10.14 suggest " . . . as is similar to . . ."*

**Agree. Thanks.**

*10.18 " . . . having corresponding stratiform DSDs (or, the absence thereof) . . ." seems contradictory. If having a corresponding stratiform DSD is indicative of TM oceanic characteristics, why then would also its absence?*

**Agree that this is awkward. Rephrased as, "In contrast, DSDs associated with ETH < 9 km carry DSD properties most similar to TM oceanic characteristics, having corresponding stratiform DSDs that favor smaller median drop sizing than deeper column counterparts."**

*10.22 you say "adjacent" and then describe "transitional" - are they the same? and if so, can you pick one?*

**Agree. Replaced 'adjacent' with 'Transitional'.**

*11.4 "a bright band signature"*

**Agree. Thanks.**

*11.18 suggest removing "are those that"*

**Agree. Thanks.**

*12.2 Suggest excising the sentence "A more practical . . . GoAmazon2014/5." it's awkward here.*

**Agree. Have removed this line.**

*12.14 the letters in "convective available potential energy" don't need to be capitalized.*

**Agree. Thanks.**

*12.16 suggest removing "that" from "studies that indicate"*

**Agree. Thanks.**

*13.1 suggest ". . . drops, and toward parameter spaces . . ."*

**Accepted. Thanks.**

*13.3 "though they do support"*

**Agree. Thanks.**

*13.4 perhaps change title to "Role of Pollution in Oceanic Signatures"?*

**Agree. Thanks.**

*13.12 suggest "The rightmost panels of Figure 15 show a composite mean . . ."*

**Accepted. Thanks.**

*13.16 remove ", respectively"*

**Accepted. Thanks.**

*13.17-23 Here is the paragraph in which I struggled with the presentation of representative statistics.*

**Yes. We can add a line to this paragraph that points to the supplemental material that contains histograms for the MUCAPE and MUCIN.**

*14.5 "One explanation . . . is that more prominent . . . contributions are acting within these convective columns . . ."*

**Agree. Thanks.**

*14.16 " . . . suggest that cleaner aerosol conditions are associated . . ."*

**Agree. Thanks.**

*14.31 suggest " . . . wind directions, and therefore should not be as influenced . . . "*

**Accepted. Thanks.**

*14.33 suggest "(e.g., local sources)"*

**Accepted. Thanks.**

*15.19 " . . . explanations for why these outliers cluster . . ."*

**Agree. Thanks.**

*15.29 suggest " . . .initiation and subsequent precipitation . . ."*

**Accepted. Thanks.**

*15.33 change "were" to "was"*

**Agree. Thanks.**

*15.35 suggest "tended to be associated"*

**Agree. Thanks.**

*16.11 suggest removing "properties"*

**Agree. Thanks.**

*16.16-onward - use caution with throwing the word "storm" around. You haven't defined what a storm is, and I'm guessing you don't mean a thunderstorm. You don't use this word until this last section. I'd suggest saying "event" from here onward, just to be safe.*

**OK. Agree. Thanks.**

*16.22 ". . . continental behaviors as seen in previous studies . . ."*

**Agree. Thanks.**

*16.26 swap "Consulting" with "Considering"*

**Agree. Thanks.**

*16.28 remove "radiosonde" - the balloons aren't doing the forcing (although that would be some wild micro-scale meteorology!)*

**Agree. Thanks.**

*16.29 suggest "character of the congestus"*

**Agree. Thanks.**

*16.30 suggest "segregating by wind direction"*

**Agree. Thanks.**

*17.3 "a topic of future consideration"*

**Agree. Thanks.**